# Single-molecular insights into the breakpoint of cellulose nanofibers assembly during saccharification

Ran Zhang [1,2,3,4,8], Zhen Hu[1,2,3,5,8], Yanting Wang[1,2,3], Huizhen Hu[1], Fengcheng Li[1], Mi Li[6], Arthur Ragauskas [6], Tao Xia[1,3,7], Heyou Han[7], Jingfeng Tang[2], Haizhong Yu [3] ✉, Bingqian Xu [4] ✉ & Liangcai Peng [1,2,3] ✉

Plant cellulose microfibrils are increasingly employed to produce functional nanofibers and nanocrystals for biomaterials, but their catalytic formation and conversion mechanisms remain elusive. Here, we characterize length-reduced cellulose nanofibers assembly in situ accounting for the high density of amorphous cellulose regions in the natural rice *fragile culm 16* (*Osfc16*) mutant defective in cellulose biosynthesis using both classic and advanced atomic force microscopy (AFM) techniques equipped with a single-molecular recognition system. By employing individual types of cellulases, we observe efficient enzymatic catalysis modes in the mutant, due to amorphous and inner-broken cellulose chains elevated as breakpoints for initiating and completing cellulose hydrolyses into higher-yield fermentable sugars. Furthermore, effective chemical catalysis mode is examined in vitro for cellulose nanofibers conversion into nanocrystals with reduced dimensions. Our study addresses how plant cellulose substrates are digestible and convertible, revealing a strategy for precise engineering of cellulose substrates toward cost-effective biofuels and high-quality bioproducts.

Cellulose represents the most abundant terrestrial biomass resource sustainable for conversion into biofuels and bioproducts[1,2]. As a major cell wall polymer and great carbon sink for land plants, cellulose is composed of β−1,4-glucan chains aligned that form crystalline microfibrils via hydrogen bonds and intermolecular forces[3,4]. As native intact elements bound to plant cell walls[5], cellulose microfibrils provide both strength and flexibility to plant cells and exhibit a recalcitrant property against biotic stress and enzymatic hydrolysis[6]. This recalcitrance contributes to the cost of cellulose conversion for biomass saccharification and bioproduction at large scale[7].

Cellulose in higher plants is synthesized by cellulose synthase complexes on plasma membranes of plant cells[8]. In the genetics-model rice, three cellulose synthase isoforms (CESA4, CESA7, CESA9) are essential for the cellulose synthesis of secondary cell walls[9,10]. Despite the super-macromolecular structure of CESA complexes favor for clustering of β−1,4-glucan chains to form crystalline microfibrils, dynamic cell wall deposition demands amending cellulose microfibrils

[1]Biomass & Bioenergy Research Centre, College of Plant Science & Technology, Huazhong Agricultural University, Wuhan 430070, China. [2]Key Laboratory of Fermentation Engineering (Ministry of Education), College of Biotechnology & Food Science, Hubei University of Technology, Wuhan 430068, China. [3]Laboratory of Biomass Engineering & Nanomaterial Application in Automobiles, College of Food Science & Chemical Engineering, Hubei University of Arts & Science, Xiangyang 441003, China. [4]Single Molecule Study Laboratory, College of Engineering, University of Georgia, Athens, GA 30602, USA. [5]College of Resources & Environment, Huazhong Agricultural University, Wuhan 430070, China. [6]Department of Chemical & Biomolecular Engineering, University of Tennessee-Knoxville, Knoxville, TN 37996, USA. [7]State Key Laboratory of Agricultural Microbiology, College of Life Science & Technology, Huazhong Agricultural University, Wuhan 430070, China. [8]These authors contributed equally: Ran Zhang, Zhen Hu. ✉e-mail: haizhongvip@hbuas.edu.cn; bxu@engr.uga.edu; lpeng@mail.hzau.edu.cn

orientation, which is assumed to be a major cause for amorphous/non-crystalline cellulose formation[11,12]. The small-angle neutron scattering result implicates a periodical distribution of the amorphous cellulose regions with an average of 300 glucose residues intervals, accounting for the crystalline units at ~150 nm length[13]. However, recent NMR studies suggest that the amorphous cellulose may embrace the crystalline cores of microfibrils[14]. Since the amorphous cellulose region is susceptible to enzymatic and acidic hydrolyses[15], it is a high priority to explore its role during the hydrolysis and conversion of intact cellulose microfibrils. In addition, it remains to investigate the amorphous cellulose's role in producing cellulose nanocrystals, the dispersed spindle-shaped nanoparticles for high-value nanomaterials[16].

Cellulose synthesis mutants are ideal genetic resources to study cellulose structural variation due to the capacity of tailoring cellulose structure compared with wild type (WT) plants. A previously reported rice mutant *fragile culm 16* (*Osfc16*) has reduced degree of polymerization (DP) of cellulose chains and distinct cellulose microfibrils assembly[17,18]. Compared with its WT, the *Osfc16* mutant increases hexose yield by 2.3-fold after chemical (acid, alkali) pretreatments of stem tissue[17], and generates distinct cellulose nanofibers with much improved Pickering emulsions productivity and lignocellulose-degradation enzymes[18].

In this work, we sequentially extract lignin and hemicellulose from the stem tissues of rice *Osfc16* mutant and its WT to expose native cellulose nanofibers distribution in situ, and probe the cellulose nanofibers length by scaling the average distance of two amorphous cellulose regions on the surfaces of cellulose microfibrils. By integrating the classic and advanced atomic force microscopy (AFM) techniques, we estimate the length-reduced cellulose nanofibers in the *Osfc16* mutant. Furthermore, employing individual types of cellulases, we observe inner-broken cellulose nanofibers distribution from distinct time-course enzymatic digestions of individual cellulose microfibrils in the mutant. Finally, we detect significantly raised fermentable sugar yield and size-reduced cellulose nanocrystals in the *Osfc16* mutant, thereby revealing enzymatic and chemical catalysis modes from distinct cellulose nanofibers assembly.

## Results

### Single-molecular recognition into cellulose nanofibers assembly
To explore native cellulose microfibril's ultrastructure, we initially applied the classic AFM technique to observe the delignified cell walls of rice stem tissues in situ at the heading stage (Fig. 1a)[19,20]. Using the previously identified rice mutant *Osfc16* and its WT (*Oryza sativa* L. ssp. *Japonica* cv. Nipponbare/NPB)[17], we attempted to consecutively remove lignin from cellulose microfibrils with acidic chlorite at a series of concentrations (4–32% NaClO$_2$) under low temperature (Fig. 1b; Supplementary Fig. 1). These NaClO$_2$ treatments removed >90% lignin and <30% hemicellulose, and the Fourier transform infrared (FT-IR) spectra implicated an obvious reduction of the three major chemical bonds involved in lignin interlinkages (Supplementary Fig. 1a–d and Supplementary Table 1). Meanwhile, X-ray diffraction (XRD) scanning exhibited a similar spectroscopic pattern among all examined samples (Supplementary Fig. 1e, f), suggesting that the NaClO$_2$ treatments did not significantly alter the native crystalline polymorphic state of cellulose microfibrils. In addition, the cellulose crystallinity index (CrI) values were elevated in the delignified samples, which were attributed to the coextraction of lignin and hemicellulose from the plant cell walls.

Employing the optimal delignification process with 8% NaClO$_2$, the native microfibrils assembly was observed in the plant cell walls of the stem tissue at the heading stage (Fig. 1b). The WT exhibited a typical microfibrils-crossed orientation pattern as reported in other plant species[5,19,20]. In contrast, the microfibrils are tangled at a small angle in the *Osfc16* mutant, probably due to its reduced degree of polymerization (DP) of β−1,4-glucans detected by two independent

assay methods (Fig. 1c, d)[17]. While further treated with a higher concentration of NaClO$_2$, significant defective/broken points appeared along the cellulose microfibrils' surface, creating discontinuous nanofibers in both the mutant and WT (Fig. 1b). Therefore, the cellulose nanofibers were measured by scaling the AFM topography images that implicate the average distance of two defects derived from the possible amorphous cellulose regions from 16% NaClO$_2$ treatment (Fig. 1e). The *Osfc16* mutant had length-reduced nanofibers of 61 nm compared with the WT of 97 nm (Fig. 1f), which is an evidence of an increased density of amorphous cellulose regions in the mutant. Given that our previously identified rice mutants (*Osfc9* and *Osfc24*) could expose cellulose nanofibers from 8% NaClO$_2$ extractions (Supplementary Fig. 2)[20,21], we assumed that the defects of whole CESA4, 7, 9 complexes at both mutants might cause a relatively easy extraction of cellulose nanofibers. By comparison, the *Osfc16* mutant with the site-mutation of only CESA9 isoform, presumably less defective for cellulose biosynthesis, which thus requires the 16% NaClO$_2$ extraction for in situ observation of cellulose nanofibers.

To corroborate this finding, an advanced AFM technique was established by integrating a single molecular recognition imaging system via the CBM3a probe, a carbohydrate-binding module specific for binding to crystalline cellulose (Fig. 2a)[22,23]. This advanced methodology enables us to observe topography and specifically recognized images of the crystalline cellulose microfibrils simultaneously[22,23]. In this study, it was applied to explore the distinct distribution of the crystalline cellulose microfibrils for the *Osfc16* mutant and WT. To assure recognition specificity, control experiments were conducted to test the CBM3a probe for blank samples (buffer or glass plate only) or untreated cell walls (without NaClO$_2$ extraction), as well as the non-CBM3a probe with cellulose microfibrils. As a result, the control experiments showed low recognition area (dark regions in recognition image) and non-specific CBM3a binding of most regions (Fig. 2b–d). By comparison, we observed high and matched recognition area and detected specific binding by monitoring the rupture force between the crystalline cellulose microfibrils and the CBM3a (Fig. 2e). Based on Bell's single barrier model and the Jarzynski equation[24,25], the CBM3a probe was evaluated with a specific and consistent binding affinity with crystalline cellulose microfibrils (Fig. 2f–j). Hence, the AFM topography and recognition by CBM3a were imaged successfully (Fig. 3a, b). The recognized cellulose nanofiber lengths were also estimated to account for the density of amorphous cellulose regions in the *Osfc16* mutant and WT (Fig. 3c–f). Notably, similar cellulose nanofibers lengths of the *Osfc16* mutant and the WT were assessed between the classic AFM and CBM3a-probed AFM images, implicating that the two AFM approaches are capable of in situ identifying cellulose nanofibers in plant cell walls. In addition, the periodical distributions of cellulose nanofibers with different frequencies were observed between the *Osfc16* mutant and WT.

### Single-enzyme catalysis modes for cellulose nanofibers hydrolyses
As the cellulose nanofibers originated from two proximate defects derived from the amorphous cellulose regions, this study attempted to test their roles assumed for initial enzymatic catalysis. Hence, we performed in situ time-course enzymatic hydrolyses in delignified stem tissues of *Osfc16* mutant and WT by adding individual endoglucanase (EG) or a mixture of cellobiohydrolase (CBH) and glucosidase (BG). The enzymes are considered as three major cellulases essential for cellulose hydrolysis into glucose[26,27]. In line with the assumption that EG initiates the hydrolysis of amorphous cellulose to release mono- and oligosaccharides[28,29], we observed a constant and slow EG digestion to break the microfibrils into independent nanofibers after a 40 min incubation in WT. Its crystalline regions remained intact (Fig. 4a). By comparison, the nanofibers from the *Osfc16* mutant partially split at the initial hydrolysis time. Prolonged incubation with EG

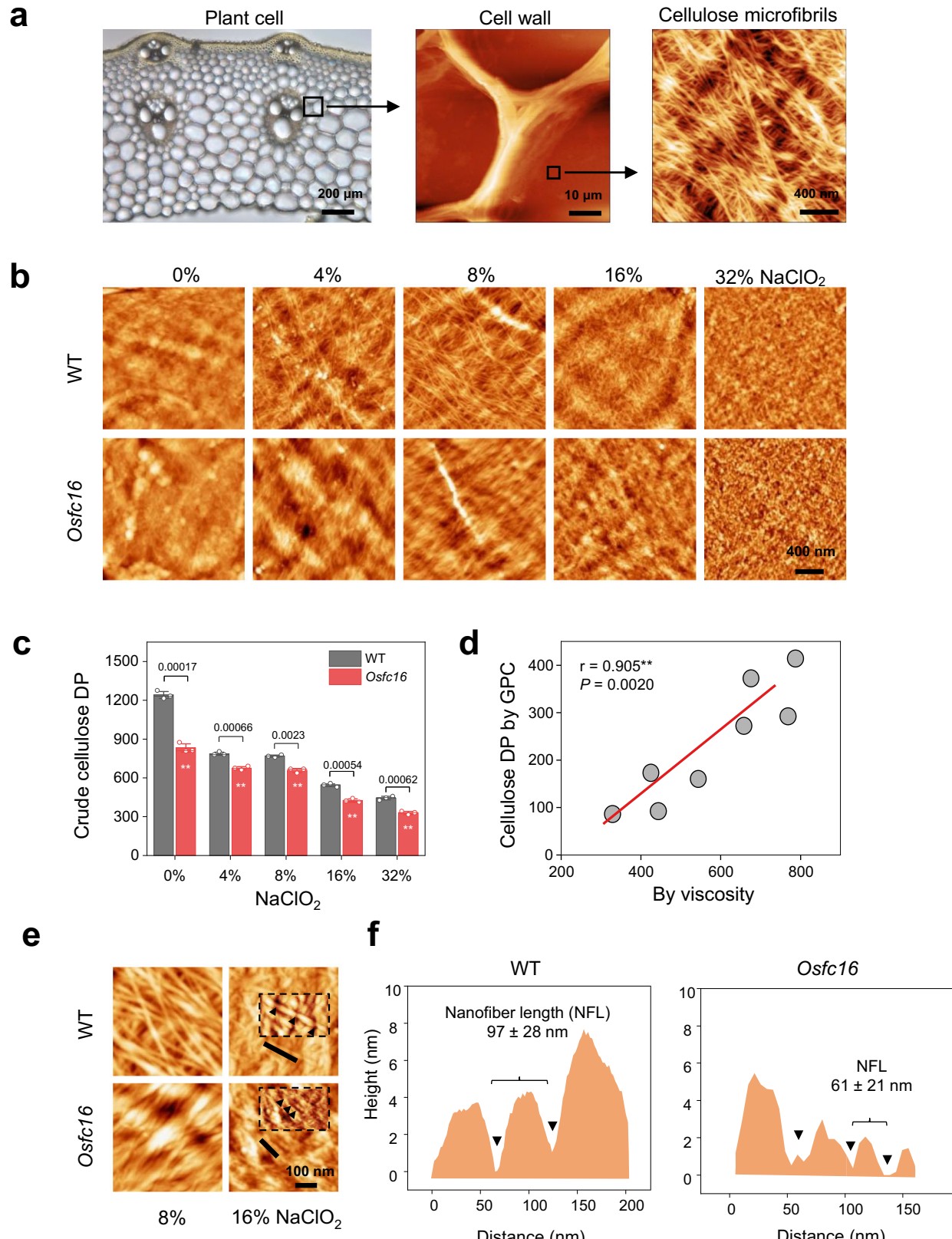

led to shorter-nanofibers assembly until the inner cellulose microfibrils of plant cell walls were observed after 210 min incubation. While incubated with mixed-enzymes of CBHI and BG, the entire cellulose microfibrils of WT were getting thinner slowly until small-size nanofibers were visible after 78 min (Fig. 4b), providing direct evidence that the CBH enzyme could catalyze the peeling off of entire β−1,4-glucan

chains starting from the reducing ends (breakpoints) of the microfibrils to release cellobiose for BG digestion into glucose. This observation is consistent with the previous findings about the sliding movement of CBH along cellulose for its processive degradation[30]. Moreover, considering that CBH requires initial recognition of cellulose chains, our findings of the breakpoints intervals along cellulose

**Fig. 1 | In situ measurements of cellulose microfibrils and cellulose nanofibers in WT and *Osfc16* mutant stem tissues. a** Dissection of plant cell walls for observation of cellulose microfibrils (CMFs) in situ in rice stem tissues in this study. From left to right, optical microscope view of plant cells in stem crosscutting slices, AFM observation of the cell wall, and AFM image of microfibrils in the innermost surface of plant cell wall. **b** Classic AFM topography of CMFs and cellulose nanofibers (CNFs) in plant cell walls after NaClO$_2$ treatments at different concentrations. **c** Crude cellulose DP values by viscosity assay. Bar as means ± SD (*n* = 3 biologically independent samples); Significant differences between the WT and mutant were determined using two-tailed Student's *t*-test: **P* < 0.01. **d** Correlation analysis for

cellulose DP by viscosity and GPC methods. ** As significant correlation at *P* < 0.01 level by two-tailed Spearman's method (*n* = 8 data pairs by two DP measuring methods). **e** Classic AFM topography images highlighting breakpoints of CNFs after 16% NaClO$_2$ treatments. Black arrows in the higher magnification inset panels indicate the breakpoints. **f** Cross profiles corresponding for black lines in **e** to illustrate alternate breakpoints for nanofibers lengths. NFL, nanofibers length, data as means ± SD (*n* = 100 nanofibers counted from three biologically independent samples). AFM experiments were repeated at least three times independently with similar results. Source data are provided as a Source Data file.

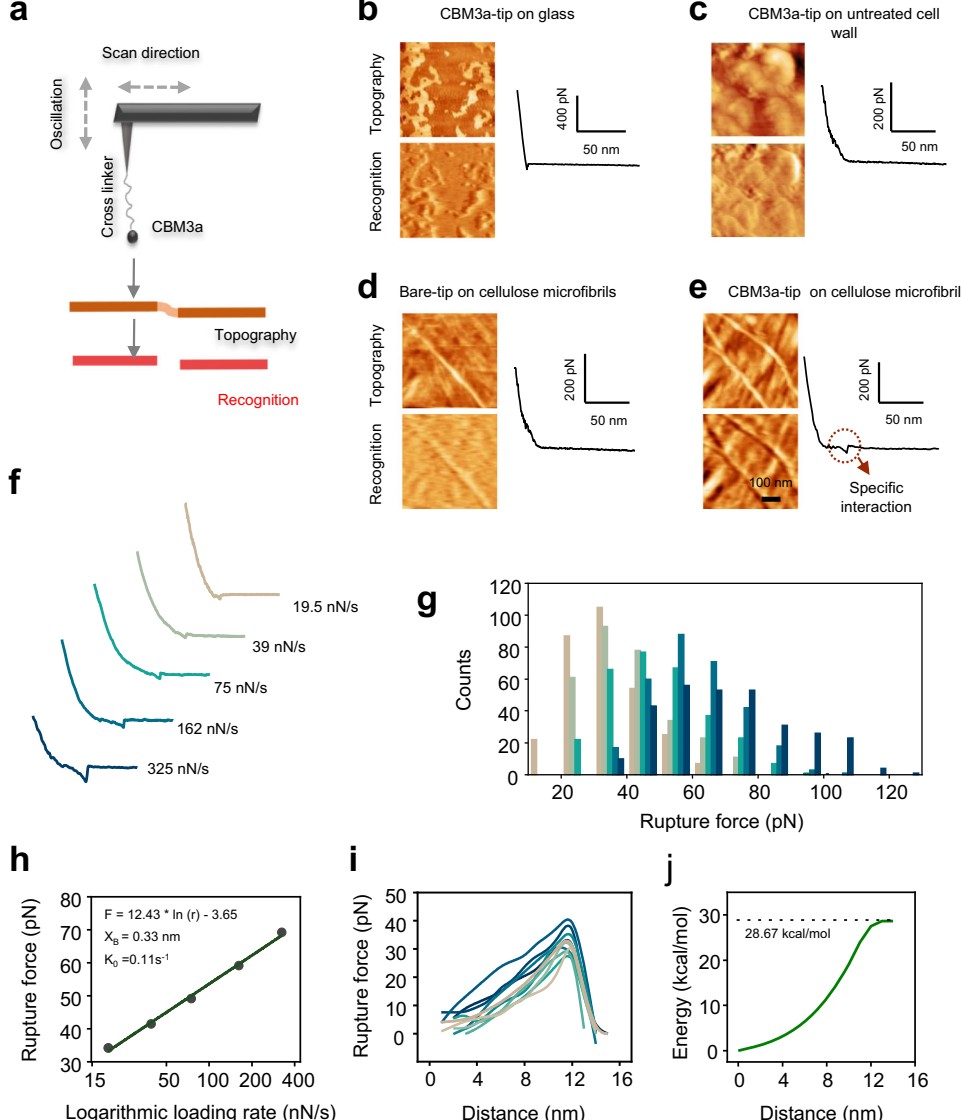

**Fig. 2 | Recognition of crystalline cellulose with CBM3a-probe. a** Schematics of the CBM3a-probed AFM specific for binding with crystalline CMFs. **b–e** Control experiments for CBM3a interaction with CMF. **f, g** The distribution of interaction force between CBM3a and CMF at different loading rates. **h** Most probable rupture

force at different loading rates, subjective to the Bell model. **i, j** The average curve of the force spectrum calculated for the free energy of interaction between CBM3a and CMFs. Source data are provided as a Source Data file.

suggest a coordinated and repeated stop-and-go movement of CBH molecules[31]. Since the *Osfc16* mutant has more amorphous cellulose regions, the mixed-enzymes (CBHI + BG) could quickly access and digest amorphous cellulose regions for nanofibers assembly from the initial time, and short-length nanofibers could be completely digested to expose smooth and flat inner-faces during further enzymes incubation (Fig. 4b). These in situ real-time observations thus provide

direct evidence of the two distinct EG and CBH catalysis modes for cellulose hydrolyses (Fig. 4c).

Furthermore, classic and single-molecular AFM approaches were applied to observe the assembly of nanofibers in the EG-digested samples (Fig. 4d, e). As a result, cellulose nanofibers lengths from WT had comparable lengths of 96 and 92 nm measured by the two AFM approaches respectively, but the *Osfc16* mutant showed varied

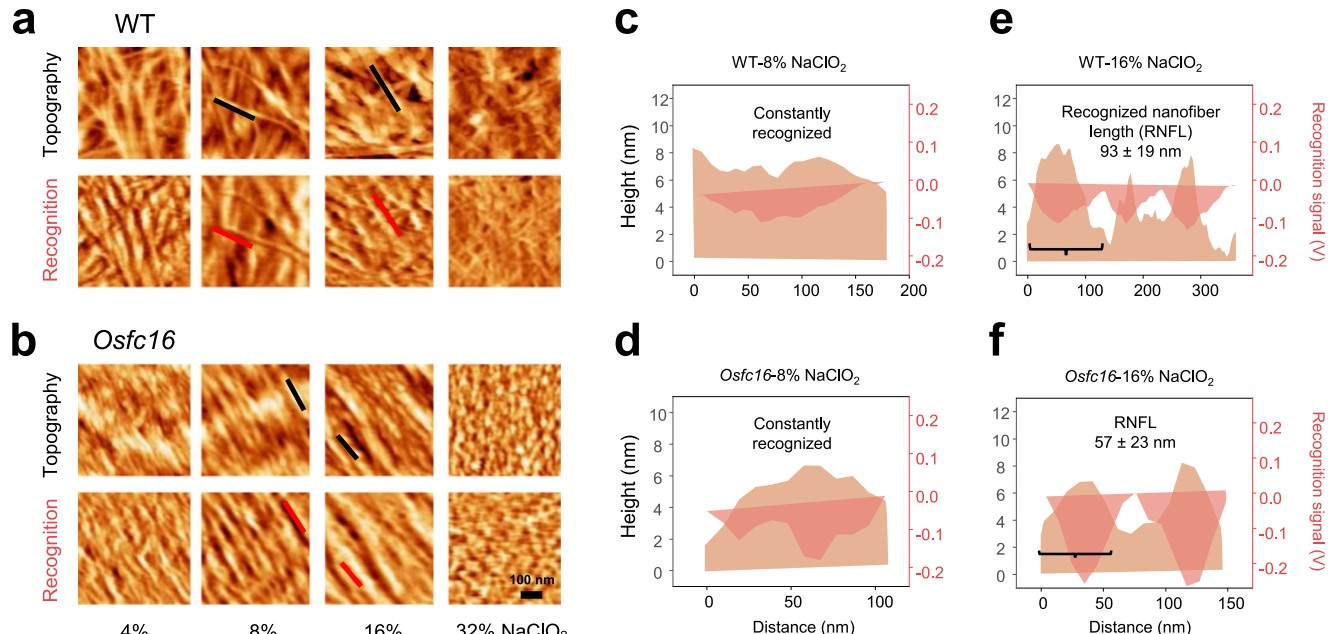

**Fig. 3 | Recognition of CMFs and CNFs in WT and *Osfc16* mutant. a, b** CBM3a-probed AFM topography and corresponding recognition images after NaClO₂ treatments at different concentrations. Black and red lines indicate fibers in topography and recognition images, respectively. **c, d** Cross profile corresponding for black and red lines to recognize CMFs (8% NaClO₂). **e, f** Cross profile

corresponding for black and red lines to recognize CNFs. RNFL, recognition nanofibers length (16% NaClO₂); Data as means ± SD ($n = 100$ recognized nanofibers counted from three biologically independent samples). AFM experiments were repeated at least three times independently with similar results. Source data are provided as a Source Data file.

nanofibers lengtat 50 and 41 nm (Fig. 4f, g). Further, compared with the 16% NaClO₂ extracted nanofibers at 57 nm in the *Osfc16* mutant by the single-molecular AFM (Fig. 3f), the EG digestion of 8% NaClO₂ extracted *Osfc16* sample led to a nanofiber length of 41 nm, indicating that the EG digestion was more specific for digestion of amorphous cellulose regions (Fig. 4g). However, the nanofiber with similar lengths (92, 93 nm) was found in the WT by the EG-digested and 16% NaClO₂-extracted methods (Figs. 3e, 4g), suggesting that the single-molecular AFM may be more sensitive for measurement of short cellulose nanofibers which occurred in the mutant. Hence, the results showed that native amorphous cellulose regions act as the breakpoints for nanofibers assembly from enzymatic digestion or chemical extraction.

To further confirm the distinct cellulose nanofibers assembly, we measured the pore size and distribution of intact cellulose microfibrils using the N₂ adsorption/desorption method[32]. Compared with the WT, the *Osfc16* mutant showed increased number of micropores (2–50 nm) and nanopores (<2 nm) in the cellulose microfibrils (Supplementary Fig. 3), which also accounts for the more cellulose breaking regions in the microfibrils of *Osfc16* mutant as assumed in the catalysis mode (Fig. 4c).

**Commonly enhanced biomass enzymatic saccharification**
To further explore the in vitro roles of the amorphous cellulose in biomass enzymatic saccharification, mature rice straw samples treated with different NaClO₂ concentrations were incubated with several formulas of cellulases (EG, CBHI + BG, EG + CBHI + BG, and two commercial mixed-cellulases: CTec2, HSB) to measure the hexose released from the enzymatic hydrolyses (Fig. 5a–e; Supplementary Fig. 4). In general, both the *Osfc16* mutant and WT produced the highest hexose yields from the 8% NaClO₂ extracted samples, with near-complete saccharification of cellulose in the mutant quantified by GC/MS analysis (Fig. 5f). Noteworthy, almost all *Osfc16* mutant samples exhibited much higher hexose yields than those of WT at $P < 0.01$ levels ($n = 3$), which could be attributed to its raised amorphous cellulose regions as breakpoints for initiating and completing enzymatic hydrolysis of cellulose nanofibers. Nevertheless, further NaClO₂ treatments at high

concentrations (i.e., 16 and 32%) led to reduced hexose yields for both mutant and WT samples, probably due to the partial removal of the amorphous cellulose regions from the fibers with increased NaClO₂ concentration.

Furthermore, we studied the enzymatic hydrolyses of two distinct substrates of six major bioenergy crops such as rice (*Osfc16*, WT), wheat, corn, *Miscanthus*, and poplar (Fig. 5g–i): the whole cellulose microfibrils (from 8% NaClO₂ extraction at 50 °C) and the crystalline cellulose substrates (from nitric-acetic acids extraction at 100 °C). We found that the whole cellulose microfibrils released much higher hexose than those of the crystalline cellulose substrates up to 2–10 folds. The lower hexose hydrolysis yields of all the six biomass samples were attributed to the removal of amorphous cellulose regions from the nitric-acetic acids treatment[33]. This finding supports our hypothesis that the native amorphous cellulose regions are essential breakpoints for initiating and completing enzymatic catalysis of biomass saccharification in bioenergy crops.

**Characteristic chemical catalysis modes for cellulose nanocrystals production**
Concerning the amorphous cellulose regions as breakpoints for cellulose nanofibers assembly, we treated the cellulose microfibrils with 64% H₂SO₄ (w/w, 45 °C) to generate completely separated cellulose nanocrystals in vitro, which has been considered as the optimal intermediates for highly valuable bio-derived nanocellulose[34]. The H₂SO₄ treatment is a well-established approach to efficiently cleave off all surface amorphous and inner-broken cellulose chains from the entire cellulose microfibrils[34], affording spindle-like cellulose nanocrystals for all samples observed under AFM (Fig. 6a) and quantified in terms of the average lengths and diameters (Fig. 6b, c). Compared with the nanocrystals from WT with a length of 218 nm and a diameter of 8.1 nm, the *Osfc16* mutants produced smaller nanocrystals with reduced lengths by 26% and diameters by 25%, consistent with its length-reduced cellulose nanofibers assembly. Therefore, the results also suggest that the high density of amorphous cellulose regions and much inner-broken cellulose chains are essential for the generation of

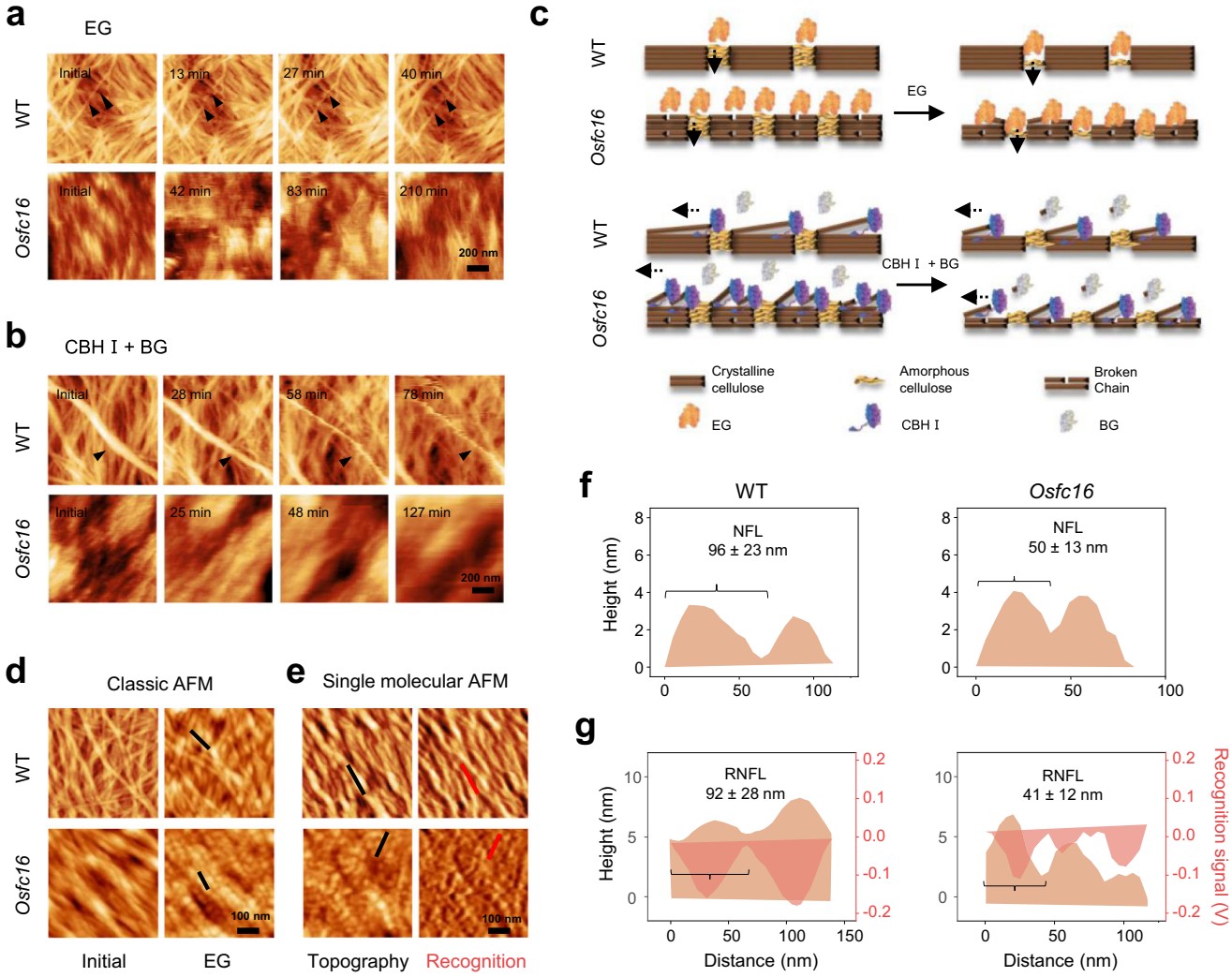

**Fig. 4 | In situ enzymatic hydrolyses of CMFs into CNFs by real-time and recognition AFM observation in WT and *Osfc16* mutant. a, b** AFM topography images of CMFs (8% NaClO₂ extraction) after time-course enzymatic hydrolyses by EG or mixed-cellulases (CBHI + BG), preparing samples for (**d, e**). **c** Schematic illustration of distinct catalysis modes for cellulose hydrolyses with EG and mixed-cellulases (CBHI + BG), respectively. **d, e** Classic and single-molecular AFM images for nanofibers from the samples in **a. f, g** Cross profile corresponding for lines in

**d, e** to recognize CNFs produced by EG hydrolyses. NFL, nanofibers length; RNFL, recognition nanofibers length; Data show average nanofibers lengths from AFM images in **f, g** as means ± SD (*n* = 100 nanofibers or recognized nanofibers counted from three biologically independent samples). AFM experiments were repeated at least three times independently with similar results. Source data are provided as a Source Data file.

nanocrystals with reduced length and diameter, respectively, in a chemical catalysis mode (Fig. 6c).

## Discussion

Although plant cellulose microfibrils have been broadly applied to generate functional substrates and intermediates for biofuels and bioproduction[7], lignocellulose recalcitrance requires a costly conversion[35]. Hence, sorting out the molecular mechanisms of enzymatic and chemical catalysis is essential to convert whole cellulose microfibrils effectively. However, it remains technically challenging to delineate the complex ultrastructure and diverse functions of cellulose microfibrils[3,4]. By integrating chemical and enzymatic dissections of native cellulose microfibrils, we demonstrated that both classic and single-molecular-probe AFMs are applicable to in situ probe amorphous cellulose regions that scale cellulose nanofibers assembly in plant cell walls of rice *Osfc16* mutant and its WT. This study not only provides the methodology for real-time observation of single-enzyme catalysis for native cellulose microfibrils hydrolysis, but also sorts out amorphous cellulose regions as the breakpoints for initiating and

completing cellulose nanofibers saccharification into fermentable sugars readily convertible for biofuels and biochemicals, which sheds light on how plant cellulose substrates are digested and converted. Therefore, this study could provide distinct enzymatic catalysis modes for near-complete biomass enzymatic saccharification even under mild biomass pretreatments for desirable bioenergy crops.

As cellulose synthase complexes are precisely constructed to produce native cellulose microfibrils in plant cell walls[4], how amorphous cellulose formation during plant growth and development is still under debate. The origin of amorphous cellulose formation has several physical and chemical explanations, such as twisting and bending microfibrils and converging and diverging bound hemicelluloses[19,36–38]. By measuring either reduced cellulose nanofibers length in situ or reduced DP value of β−1,4-glucans in vitro in the previously identified rice *Osfc16/Oscesa9* mutant[17,18], the results indicate that cellulose biosynthesis might be one of the major causes for forming amorphous cellulose in plant cell walls, which could thus explicate our previously identified two rice mutants (*Osfc9* and *Osfc24*) that are of defective cellulose biosynthesis in secondary cell walls of

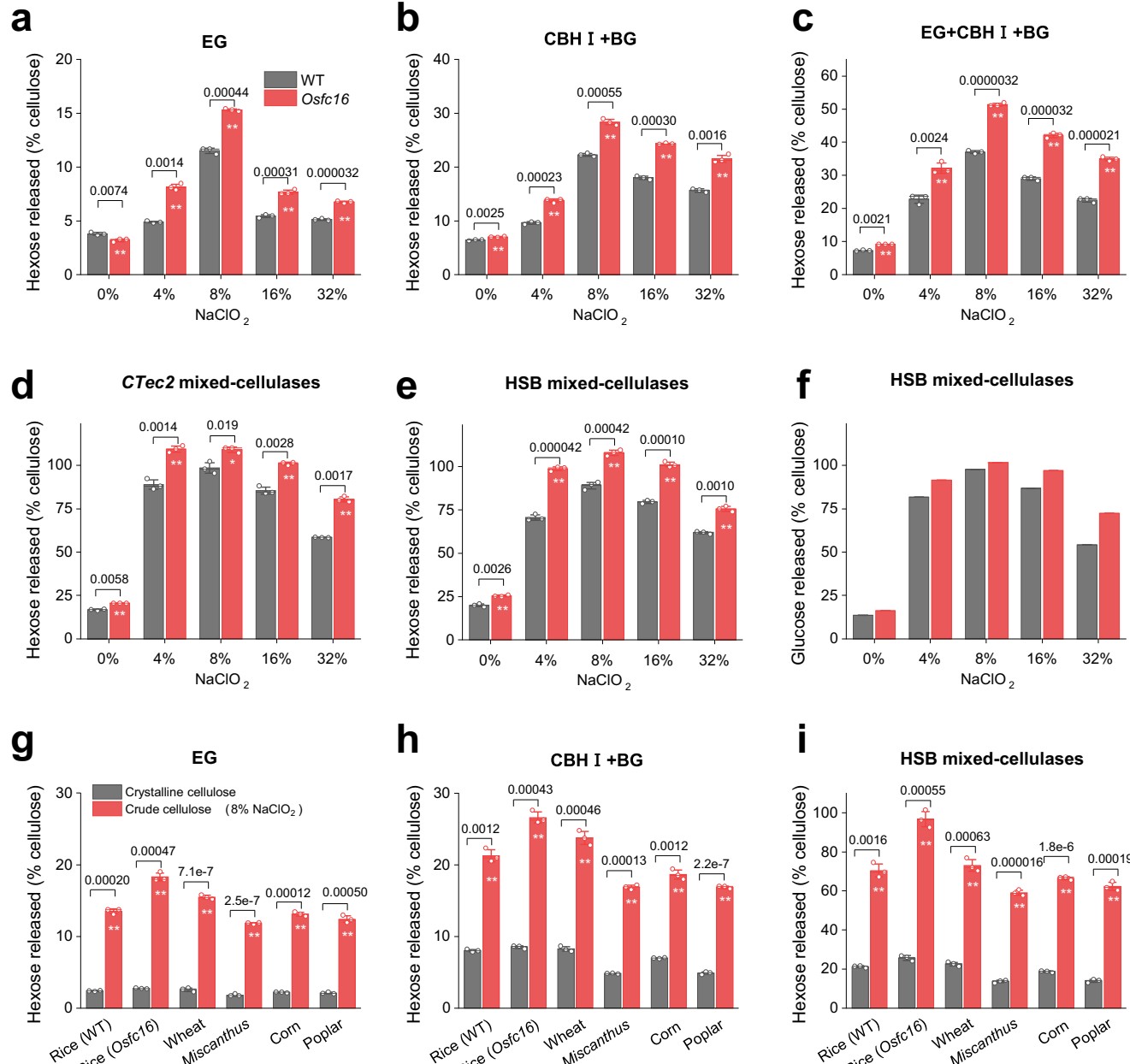

**Fig. 5 | CMFs integrity and CNF length for enzymatic saccharification.**
**a**–**e** Measurement of hexose (glucose) yields (% cellulose) released from enzymatic hydrolyses with de-lignin powders samples of *Osfc16* mutant and WT in vitro. Hexose yields by different formulas of enzymes with single cellulase or commercial mixed-cellulases. **f** Glucose yields by GC/MS analysis of hexose in **e**, data extracted from GC/MS spectroscopic profiling. **g**–**i** Hexose yields released from enzymatic hydrolyses of crude cellulose (8% NaClO$_2$ extraction) and crystalline cellulose substrates in five major bioenergy crops. Different formulas of enzymes with single cellulase or commercial mixed-cellulases. Data with error bars as means ± SD ($n = 3$ biologically independent samples). Significant differences in each group were determined using two-tailed Student's $t$-test: \*\**P* < 0.01, \**P* < 0.05. Source data are provided as a Source Data file.

stem tissues[20,21]. Notably, we could observe cellulose defects of those two mutants under 8% NaClO$_2$ extraction (Supplementary Fig. 2), suggesting that cellulose defects should also depend on genetic engineering of cellulose biosynthesis.

Furthermore, our other study reports that OsCESA4, 7, 9 isoforms could be equally functional for the biosynthesis of β−1,4-glucan chains to form cellulose microfibrils in plant cell walls, and three isoforms are likely to form heterotrimeric structure with symmetric CESA distribution for cellulose synthase complexes[39]. Hence, the site-mutation of OsCESA9 should cause a symmetric distribution of DP-reduced cellulose chains from the early termination of β−1,4-glucan chain elongation in the *Osfc16* mutant[17], which explains why the *Osfc16* mutant has more raised amorphous cellulose regions and inner-broken chains occurrence of

whole cellulose microfibrils. In addition, the rice *Osfc16* site mutant has normal growth and improved lodging resistance despite its defects in cellulose biosynthesis[17]. Using the CRISPR/Cas9 gene-editing method, we create another site mutant of OsCESA9 that shows length-reduced cellulose nanofibers in situ and higher enzymatic saccharification in vitro[40], consistent with the findings of the *Osfc16* mutant examined in this study[17]. Moreover, this *cesa9* mutant generates high-porosity bio-char for raised dye adsorption[40]. Therefore, ideal *cesa* mutants could be further selected from specific site-mutations of major *CesAs* genes in major bioenergy crops by performing CRISPR/Cas9 gene-editing technology in the future[41].

As this study observed the surface amorphous cellulose regions as the breakpoints for initiating chemical catalysis to produce cellulose

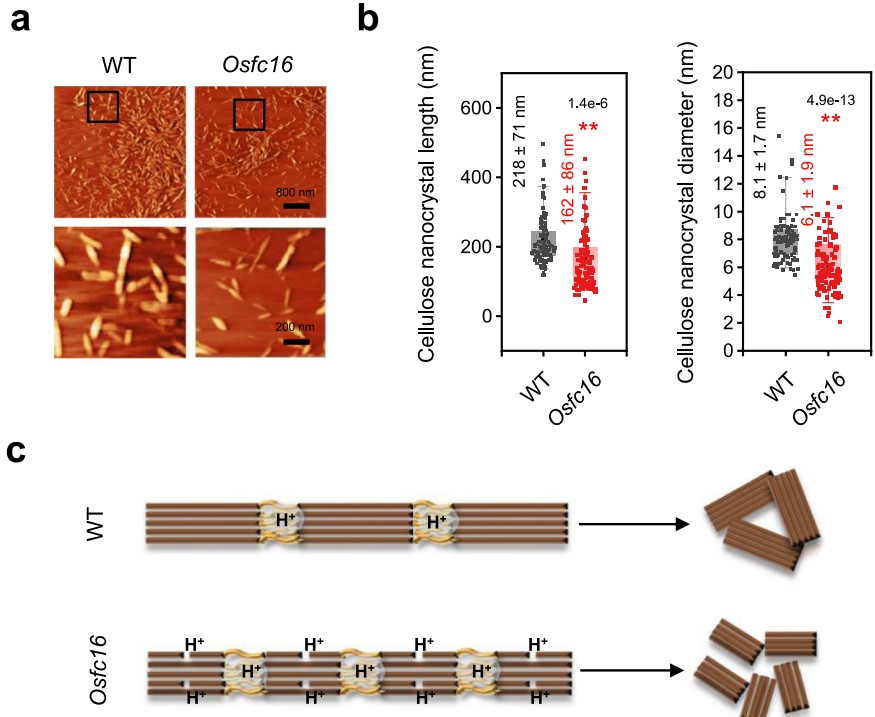

**Fig. 6 | Smaller size cellulose nanocrystals and raised nanopore volumes in *Osfc16* mutant. a** AFM observation of cellulose nanocrystals. The areas in the black boxes are magnified in the lower images. **b** Measurement of average length and diameter of nanocrystals by randomly selecting 100 nanocrystals from three biologically independent samples. **c** Acidic catalysis modes for distinct cellulose nanocrystals. Data in **b** are displayed as box and whisker plots with individual data points. The error bars represent the 95th and 5th percentiles. Centerline, average; box limits, 25th and 75th percentiles. Significant differences between the WT and mutant were determined using two-tailed Student's *t*-test: **$P < 0.01$, *$P < 0.05$. AFM experiments were repeated at least three times independently with similar results. Source data are provided as a Source Data file.

nanocrystals, the inner-broken cellulose chains are also presumed in the *Osfc16* mutant, leading to effective chemical catalysis mode for generating size-reduced cellulose nanocrystals. Interestingly, as this study found that the average cellulose nanofibers lengths are much shorter than those of the cellulose nanocrystals examined, we suppose that the cellulose nanofibers length is reduced by the relatively raised density of amorphous cellulose regions on the surfaces of whole cellulose microfibrils, which should be determined by both cellulose biosynthesis and hemicellulose interactions. By comparison, the strong 64% $H_2SO_4$ process removes surface cellulose and degrades small-size nanocrystals, leading to the cellulose nanocrystals forming mainly from the inner long cellulose chains. On the other hand, the results also indicate that the presence of several inner-broken cellulose chains could be the reason of why the *Osfc16* mutant has shorter nanocrystals than that of the WT.

Consequently, genetic selection of the ideal *cesa* mutants should further reduce cellulose nanofibers lengths to produce the optimal cellulose nanocrystals with high surface-area-to-volume ratios and reducing ends for chemical reactions that should be studied in the future[42,43]. As the cellulose nanofibers of *Osfc16* mutant have been applied as an inducing substrate for *T. reesei* to secret cellulase enzymes at high activity[18], the ideal *cesa* mutants should also be significant interest for large-scale enzyme production. This work therefore suggests a potential approach for precise genetic alteration of cellulose nanofibers along with efficient enzymatic and chemical catalysis for biomass saccharification and high-quality bioproduction.

## Methods
### Chemicals and reagents
Unless noted, all chemicals and reagents were purchased from Sigma-Aldrich (St. Louis, MO).

### Plant biomass collection
The wild type (WT) rice cultivar Nipponbare (NPB) and *fragile culm* (*Osfc*) mutants were used in this study. The *Osfc16* mutant was identified with a site-mutation in the P-CR conserved region of OsCESA9[17]. The *Osfc9* and *Osfc24* mutants are defective for cellulose biosynthesis in the secondary cell walls of rice stem tissues[20,21]. Rice WT and mutants were grown in the same block of a conventional paddy rice field of Huazhong Agricultural University, Wuhan, China.

Biomass materials from four plant species are used as substrates to test cellulose digestibility, which include wheat, *Miscanthus*, corn, and poplar. Mature stem tissues of these plant species were collected from different experimental fields of Huazhong Agricultural University and Hunan Agricultural University.

### Plant tissue and biomass preparation
The rice second internodes were collected at the heading stage and cross-sectioned for in situ AFM observation. Internodes were embedded with 4% agar, cut into half of the cross-sections 80 μm thick by a microtome (VT1000S, Leica), and suspended in ultrahigh purity water. The slices were immobilized on poly-L-lysine-coated glass slides in a vacuum. To functionalized glass slides, the glass slides were washed with acid alcohol (1% HCl in 70% ethanol) in ultra-wave, coated with a dilute poly-L-lysine solution (P8920; Sigma; 1:10 with deionized water) for 5 min, and baked in a 60 °C oven for 1 h. The immobilized slices were checked with a bright field light microscope, and the intact slices that exposed the innermost surface of the parenchyma secondary cell wall were selected for AFM observation.

The mature stem tissues of different plant species were dried at 55 °C, cut into small pieces, ground through a 40-mesh screen (0.425 × 0.425 mm), and stored in a dry container as biomass powders samples. They were used for cell wall fraction, CNC production, and

cell wall feature analysis, including CrI, DP, FT-IR, porosity, and biomass digestibility.

## Cellulose microfibrils preparation

Acid chlorite treatments were applied for lignin removal to expose cellulose microfibrils in situ by mixing 1 mM HCl with 1 g sodium chlorite (NaClO$_2$)[19,20]. The stem tissue section slices or biomass powders were gradually incubated with rising concentrations of acid sodium chlorite (0%, 4%, 8%, 16%, and 32%, w/v) at 50 °C with 2 cycles (24 h per cycle). The chlorite solution was sufficiently incubated with the section slices, whereas the chlorite was added into the biomass powders at 50:1 (v/w) proportion. The samples were washed with ultrahigh purity water for each treatment cycle until pH 7.0. The slices samples were stored in a vacuum for AFM observation, and the powders samples were washed twice with pure methanol and anhydrous acetone, dehydrated in the hood overnight and dried in the oven at 50 °C for 2 h. The dried biomass powders were ground through a 40-mesh screen and stored in a dry container until use.

## AFM observation

The AFM tips (CS-25 silicon, Lot: AP50152) with a nominal spring constant of about 0.1 N/m were purchased from Nanoscience Instruments, Phoenix, AZ. The His$_6$-tagged CBM3a was purchased from the Plant Probe (University of Leeds, UK). The HS-PEG2000-NTA crosslinker (Lot: JG125493, Nanocs Inc, NY) was applied for sufficient CBM3a interactions with crystalline cellulose microfibrils in the slice samples.

The AFM tip functionalization with CBM3a was used to recognize crystalline cellulose specifically[22,23]. The nickel and gold-coated AFM tips were incubated in the HS-PEG2000-NTA crosslinker (0.2 mg/mL, 400 µL) for 4 h, and immersed in the NiCl$_2$ (10 mM, 20 µL) for 1 h at room temperature. The tips were then washed several times and incubated with 400 µL Tris-Cl buffer (10 mM Tris-Cl and 150 mM NaCl, pH 7.5) with the addition of CBM3a (27 µg/mL, 6 µL). The tips were kept at 4 °C overnight, and the modified tips were washed with Tris-Cl buffer more than three times for AFM recognition imaging.

AFM imaging was applied for the observation of cellulose microfibrils ultrastructure[20]. The PicoPlus Molecular Imaging system with a PicoScan 3000 Controller was utilized for all AFM quantitative measurements. The Agilent multifunctional AFM scanner with open-loop was used for all recognition imaging. The system was situated on the PicoPlus Isolation Chamber to avoid environmental noise. All images were obtained using non-contact, top magnetic AC (TopMAC) mode under PicoTREC (Agilent Technologies, Santa Clara, CA) with topography (height) and recognition images captured simultaneously. All samples were imaged at an average scanning speed of 1 ln/s with 512 × 512 pixels, and at least three independent trials were conducted to ensure repeatability. About 10 random zoom-in areas and 100 data points were collected for statistical analyses. PicoView (1.14) and Gwyddion (2.56) computer programs were used to collect and process AFM images.

AFM force spectroscopy was conducted to explore CBM3a interactions with crystalline cellulose microfibrils in situ[23]. While clear images were achieved, the force-distance (F-D) curves were measured at different loading rates. For each loading rate, more than 300 curves were collected to analyze the force distribution, the most probable rupture force, and the variation of stretch distance. The data analysis was achieved by PicoView (1.14) computer program. Based on Bell's model[24] and Jarzynski's equality[25], the F-D curves and unbinding forces at different loading rates were used to determine the dynamic and kinetic parameters such as energy barrier length and free energy changes.

## Enzymatic hydrolyses of plant cell walls in situ and biomass powders in vitro

The cellobiohydrolase I (CBHI, Lot #: 40203b from *Trichoderma* sp.), endo-1, 4-β-glucanase (EG, Lot #: 130501b from *A. niger*), and β-

Glucosidase (BG, Lot #: I40101b from *Thermotoga maritima*) were purchased from Megazyme (*Megazyme* International Ireland, Bray, Ireland). Two mixed-cellulases, HSB (Imperial Jade Biotechnology, Ningxia, China) and CTec2 (Novozymes, Franklinton, NC, USA), were used for biomass enzymatic hydrolyses as a parallel comparison.

For AFM imaging of real-time enzymatic hydrolysis of plant cell walls, a flow cell was used as a reaction container and AFM tip holder. The slice sample was fixed into the flow cell filled with 300 µL binding buffer (10 mM Tris-Cl and 150 mM NaCl, pH 7.5). While initial AFM images were readily obtained, the AFM scanning was immediately stopped, and the diluted enzyme solution was gently injected into the flow cell to restart AFM imaging. The EG enzyme or mixed-enzymes (CBHI + BG) were diluted to 0.28 mg/mL or 0.37 mg/mL+0.26 mg/mL using Tris-Cl buffer enzymatic, and reactions were completed at room temperature. The samples were washed with 1 mM NaOH and ultrahigh purity water to stop the reaction.

For enzymatic hydrolyses of biomass powders, the biomass samples were incubated with the mixed-cellulases (13.23 FPU/g) or EG (22 mg/g) or mixed-enzymes (CBHI at 30 mg/g and BG at 26 mg/g) co-supplied with 1% Tween-80 in 0.2 M Na-acetate buffer (pH 4.8). The sealed samples were shaken under 150 rpm at 50 °C for 48 h (mixed-cellulases) or 120 h (other enzymes). After enzymatic reactions, the supernatants were collected by centrifuging at 3000 g for 5 min to estimate total sugars (hexose and pentose) yields. The hexose yield was calculated by the following equation:

$$\text{Hexose yield}(\%) = \frac{\text{hexose released}(g)}{\text{cellulose content}(g)} \times 100 \qquad (1)$$

The results were verified by GC/MS analysis.

## Wall polymer extraction and determination

Plant cell wall fractionation was performed to analyze the composition of the biomass samples[44,45]. The biomass samples were consecutively extracted to remove soluble sugars, lipids, starch, and pectin by using potassium phosphate buffer (pH 7.0), chloroform-methanol (1:1, v/v), DMSO–water (9:1, v/v), and ammonium oxalate 0.5% (w/v). The remaining crude residues were extracted with 4 M KOH containing 1.0 mg/mL sodium borohydride for 1 h at 25 °C, and the supernatants were combined as KOH-extractable hemicelluloses fraction. The remaining pellets were applied to detect total pentoses for non-KOH-extractable hemicelluloses fraction. The total hemicelluloses level was calculated by detecting pentoses of the non-KOH-extractable pellets and total hexoses and pentoses in the KOH-extractable fraction. Crystalline cellulose level was quantified using the *Updegraff* method[33]. Colorimetric methods were applied for the determination of hexoses and pentoses[45]. Total lignin was assayed using a two-step acid hydrolysis method according to the Laboratory Analytical Procedure of the National Renewable Energy Laboratory[46]. GC/MS (Shimadzu GCMS-QP2010 Plus) method was applied to test monosaccharides released from enzymatic hydrolysis of pretreated lignocellulose[45].

## Detection of wall polymer features and biomass porosity

The viscosity and gel-permeation chromatography (GPC) methods were used to determine the degree of polymerization (DP) of cellulose samples[47,48]. The chlorite-treated biomass powders were hydrolyzed with xylanase (3.125 U/mg dry samples, pH 5.0) at 50 °C for 48 h. After being washed with distilled water until neutral pH, the remaining residues were termed crude cellulose samples for DP assay.

For the viscosity method, the sample residues were washed at least five times with distilled water until pH 7.0 and dried at 38 °C with vacuum suction filtration. The DP of crude cellulose samples was measured at 25 ± 0.5 °C using cupriethylenediamine hydroxide (Cuen) as the solvent in the Ubbelohde viscometer. The relative viscosity ($\eta_{rel}$) values were calculated using the ratio of $t/t_0$, where $t$ and $t_0$ are the

efflux times for the cellulose solution and Cuen (blank) solvent, respectively. The intrinsic viscosity was calculated by interpolation using the United States Pharmacopeia table (USP, 2002) that files the predetermined values of the product of intrinsic viscosity and concentration. The intrinsic viscosity values were converted to cellulose DP according to the equation:

$$DP^{0.905} = 0.75[\eta] \qquad (2)$$

where [η] is the intrinsic viscosity of the solution calculated by interpolation using the USP table. All experiments were carried out in biological triplicate.

For the GPC method, all samples were derivatized with phenyl isocyanate in an anhydrous pyridine system before GPC analysis. Size-exclusion separation was performed on an Agilent 1200 HPLC system (Agilent Technologies, Inc, Santa Clara, CA) equipped with Waters Styragel columns (HR1, HR2, and HR6; Waters Corporation, Milford, MA). The number-average degree of polymerization (DP$_n$) of cellulose was obtained by dividing M$_n$, 519 g/mol, the molecular weight of the tricarbanilated cellulose repeating unit:

$$M_n = \frac{\sum M_i * N_i}{\sum N_i} \qquad (3)$$

$$DP_n = \frac{M_n}{M_0} \qquad (4)$$

where the $M_n$ is the number-average molecular weight; DP$_n$ is the number-average degree of polymerization; $N_i$ is the number of moles with the molar mass of $M_i$; $M_0$ is the molecular mass of repeating unit (i.e., 519 g/mol in the case of derivatized cellulose).

The crystallinity index (CrI) of cellulose samples was detected using the X-ray diffraction (XRD) method (Rigaku-D/MAX, Ultima III; Japan)[45]. The powders samples laid on the glass holder were analyzed under plateau conditions. Ni-filtered Cu-Kα radiation (λ = 0.154056 nm) generated at a voltage of 40 kV and current of 18 mA, and the scans at a speed of 0.0197° s⁻¹ from 10° to 45° were employed to collect diffraction data for the estimation of CrI using the equation:

$$CrI(\%) = \frac{I_{200} - I_{am}}{I_{200}} \times 100 \qquad (5)$$

where $I_{200}$ is the intensity of the 200 peaks at 2θ around 22.5°, which represents both crystalline and amorphous materials, while $I_{am}$ is the minimum intensity of amorphous material between the 200 and 110 peaks at 2θ around 18°.

FT-IR spectroscopy was performed to observe chemical linkages in biomass samples using a Perkin-Elmer spectrophotometer (NEXUS 470, Thermo Fisher Scientific, Waltham, MA, USA)[49]. The biomass was finely powdered to reduce scattering losses and deformations in the absorption band. The samples were directly positioned in the path of IR light, and the spectra were recorded in absorption mode over 32 scans at a resolution of 4 cm⁻¹ in the 4000–400 cm⁻¹ region.

The cellulosic matrix's pore size distribution was analyzed using Micrometrics ASAP 2460 (USA)[32]. After initial lignin removal with 8% NaClO₂, the remaining lignocellulose samples with intact cellulose microfibrils were used for pore size determination. The micropore and nanopore volumes distribution was calculated using the Barrett-Joyner-Halenda (BJH) and Horvath-Kawazoe (HK) methods.

### Generation of cellulose nanocrystals in vitro

Cellulose nanocrystals were generated by sulfuric acid hydrolysis of crude cellulose samples[50,51]. The crude cellulose was prepared from biomass powders after initial extraction with 8% NaClO₂. The de-lignin powders were extracted five times with 5% NaOH (w/v) at 50 °C for 1 h

and washed with deionized water until pH 7.0. The samples were washed twice with pure methanol and anhydrous acetone, dehydrated in the hood overnight and dried in the oven at 50 °C for 2 h.

The acid hydrolysis was performed by soaking 0.1 g of the dry crude cellulose sample in 2 mL sulfuric acid with a concentration of 64% (w/w) at 45 °C for 1.5 h in 40 kHz ultra-wave. The acid hydrolysis was stopped by adding 10-fold chilled distilled water and centrifuging at 6000 × g for 5 min to remove any excessive acid. The remaining pellets were dialysis for 3 days using dialysis membranes with a molecular weight cutoff of 14,000 Da. After the dialysis, the solution was subjected to sonification for 15 min to avoid aggregations. The final dispersion was diluted to 0.001% (w/w) with ultrahigh purity water and shaken thoroughly for AFM observation. For CNC particle size measurement, AFM height profiles were used to assess the length and diameter of the particles[52].

### Statistical analysis

Statistical analysis was performed using IBM SPSS Statistics 26 software. Spearman's assay conducted the correlation at the two-sided 0.05 level of significance (*$P < 0.05$, **$P < 0.01$). The variation and regression analysis were completed using Origin 2018 software (Microcal Software, Northampton, MA) for the best-fit curve from the experimental data. Quantitative data are expressed as mean and the sample numbers were noted at each of the experiments. Microsoft Excel 2016 was used for the $t$-test. The experimental error was estimated by calculating the standard deviation (SD) as means ± SD.

### Reporting summary

Further information on research design is available in the Nature Portfolio Reporting Summary linked to this article.

## Data availability

Data supporting the findings of this work are available within the paper and its Supplementary Information files. A reporting summary for this Article is available as a Supplementary Information file. *Osfc16*, *Osfc9*, and *Osfc24* mutants are available from the corresponding author L.P. (lpeng@mail.hzau.edu.cn) upon request with anticipated response within two weeks. Source data are provided with this paper.

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

## Acknowledgements

We thank Professor Yun Wang for technical assistance. This work was supported by the National Natural Science Foundation of China (32170268 to L.P., 32101701 to Y.W., 32100214 to R.Z., 31571721 to T.X.), Project of Huazhong Agricultural University Independent Scientific & Technological Innovation Foundation (2662019PY054 to L.P.; 2662020ZKPY013 to Y.W.), National 111 Project of Ministry of Education of China (BP0820035 to L.P.), Project of Hubei University of Arts & Science (XKQ2018006 to Y.W.).

## Author contributions

R.Z.: methodology, investigation, visualization, software, writing-original draft; Z.H.: methodology, investigation, software, funding acquisition; Y.W., H.Hu., and F.L.: software, investigation; M.L.: investigation, writing–review; A.R.: supervision, writing–review; H.H. and J.T.: supervision; H.Y. and T.X.: supervision, funding acquisition; B.X.: methodology, supervision, writing–review, and editing; L.P.: conceptualization, supervision, methodology, writing–review and editing, project administration, funding acquisition.

## Competing interests

The authors declare no competing interests.
