## [Peer Review File · Nature Communications]

Single-molecular insights into cellulose nanofibers assembly for enzymatic and chemical catalysis of sugars and nanocrystalsReviewers' Comments:

Reviewer #1:

Remarks to the Author:

This manuscript reports research on a rice cellulose synthase mutant. The authors used AFM and other analytic methods to characterize the cell wall cellulose microfibril structure and nanocrystals generated from the rice straw, and found more amorphous regions of the cellulose microfibrils in the mutant than that in WT, which resulted in shorter nanocrystals after delignification process, and higher enzyme digestibility. This study is of interesting to understand the molecular mechanisms of biomass deconstruction process for bio-based products. There are some questions/concerns for the authors to consider.

- 1) Does the term "Single-molecular" in the title refer to cellulose? Cellulose is a bundle of beta-1,4-glucans (molecules) packed by inter- and intra-molecular interactions.
- 2) The authors use "microfibrils", "nanofibers" and "nanocrystals" for cellulose, it may be helpful to define these terms.
- 3) It seems not clear to me what is "Classic and advanced AFM" (line 27).
- 4) "high density of amorphous cellulose chains" (line 28), cellulose chain refers to beta-1,4-glucan, do the authors mean amorphous cellulose regions?
- 5) "single-molecule probe" (line 68), does the functionalized AFM tip contains only one CBM molecule?
- 6) "native cellulose microfibrils" (line 78), are the microfibrils in the delignified cell wall still "native"?
- 7) "native microfibrils ... at heating stage" (line 92-93), are these microfibrils still "native"?
- 8) "two defects/breakpoints derived from the amorphous cellulose chains" (lines 100-101), do these "defects/breakpoints" show higher or lower intensity in the AFM height images? why? how to confirm they are "amorphous cellulose chains"? not residue hemicelluloses or other matrix polymers that may be associated with cellulose surface, especially after NaClO₂ treatment.
- 9) "be accounting for an increased density of ...mutant" (lines 103-104), is the observed decrease of nanofiber length caused by genetic modification in the mutant, or a result of chemical treatment?
- 10) "two defects/breakpoints" (line 124), are these broken cellulose chains? evidence?
- 11) "reducing ends (breakpoints)" (lines 140-141), any evidence to confirm these are reducing ends in this study?
- 12) "to expose smoother and flatter surfaces" (line 145), what are those?
- 13) "native amorphous cellulose chains act... chemical extraction" (lines 153-155), what are "native amorphous cellulose chains"? do these exist in both WT and the Osfc16 mutant? why were these "amorphous cellulose chains" not detectable before treatment (Fig 3c,3d)?
- 14) "the N2 adsorption/desorption method", provide reference for this method.
- 15) "nitric-acids extraction at 100C" (line 179), it seems not relevant with this study.
- 16) "By comparison... bioenergy crops" (lines 183-187"). This conclusion statement seems not clear, if "the intact cellulose microfibrils released much higher hexose yields...why the Osfc16 mutant that appeared to be more "defects" of cellulose microfibrils were more digestible?
- 17) "with 64% H₂SO₄" (line 191), it is not very clear why these are "highly-valuable bioproduction (line 193).
- 18) "at 219.43nm length and 7.98 nm diameter" (lines 198-199), are these mean numbers? how are they correlated with Fig 3?
- 19) "inner-broken cellulose chains" (line 203), any evidence?
- 20) "for real-time observation" (line 217), any real-time data?
- 21) "cellulose biosynthesis is one of major causes for amorphous cellulose formation in plant cell walls"(lines 233-234), is the amorphous cellulose exists only in the mutant caused by site-mutant in OsCESA9?
- 22) "the inner-broken cellulose chains are also presumed in the Osfc16 mutant"(lines 243-244), are the broken chains exist in the mutant not in WT?
- 23) "reduce cellulose ... best quality in the future" (lines 243-244), why reduced cellulose nanofibers are better? references?
- 24) "Fig.1." (line 587). The area showed in Fig. 1a appeared to be parenchyma cell walls, if all AFM images were taken from this area, they were likely primary cell walls. However, the Osfc16 mutant is

a site-mutant in a secondary cell wall cellulose synthase (OsCESA9, lines 232-233), how does it affect primary cell wall cellulose synthesis?

Reviewer #2:

Remarks to the Author:

In this manuscript, the authors make use of a rice mutant with reduced cellulose DP that they had previously characterised (Osfc16) to explore the impact of cellulose DP on cellulose fiber assembly and degradation.

This manuscript requires professional editing for clarity, grammar and spelling throughout. It is hard to interpret the authors meaning in many places. Overall, the data lacks in depth analysis, particularly in relation to previous work. Strong statements are made that simply are not supported by the data.

In the introduction, some of the reviews cited are pretty old (~20 years) and I would recommend the authors identify some more recent summaries of the literature, as there has been much progress. Please remove ref#10, as there has been no subsequent evidence supporting a role for sitosterols in priming cellulose biosynthesis (in fact, this lack of evidence is discussed in reference #4).

Please introduce more information about the Osfc16 mutant and what is known about it from the previous publication. It's core to the paper so "natural mutant" "defective at [sic] cellulose biosynthesis" is not sufficient.

Fig 1e – it is challenging to see the indicated break points – I would recommend including additional higher magnification inset panels so that it is clearer for the reader.

Fig 2 – in the text you mention an "untreated cell wall" control, but the figure says lignin. I would also be surprised that you don't get any signal with untreated cell wall, as CBM3a does bind to cell wall (e.g. 10.1073/pnas.1005732107), although of course I would expect a weaker signal than in chlorite treated wall. If the data on classic AFM and CBM3a AFM is so well correlated (as the authors state), and if it also requires the chlorite treatment which removes almost all the lignin/hemicellulose, then what is the advantage?

I would have preferred to see more discussion about what is known about cellobiohydrolase and b-glucosidase activity already. For example, the Igarashi et al. 2009, 2011 papers (10.1074/jbc.M109.034611; 10.1126/science.1208386), which used AFM to look at CBH activity are not cited.

There are many statements that are simply not addressed by the data contained in this paper. For example, I don't see how you can really claim that the source of amorphous cellulose is "cellulose biosynthesis" based on the data in this paper. Yes, you have a CESA mutant with increased amorphous cellulose, but that does not indicate cause-effect, only correlation. I'm not saying it's not a possibility, I just don't see how your data supports this statement.

Abbreviations should be written out in full, as should species names e.g. line 66 should be *Oryza sativa* L. ssp japonica cv. Nipponbare (NPB); same for gene names.

There are also issues with reference formatting e.g. #3 should be O'Sullivan etc

Reviewer #3:

Remarks to the Author:

Overall a fairly well written article, the approach, the mechanism, and the results are noteworthy.

Some minor issues that authors should consider addressing to help with the clarity of the work.

1. For CNC particle size measurement, to remove any ambiguity, the authors state that they are using AFM height profiles to assess length and diameter of the particle. The authors could also consider cite the following paper or one similar to it that describes the approach used for CNC particle size measurement using AFM.

o Bushell, M., Meija, J., Chen, M., Batchelor, W., Browne, C., Cho, J. Y., ... & Johnston, L. J. (2021). Particle size distributions for cellulose nanocrystals measured by atomic force microscopy: An interlaboratory comparison. *Cellulose*, 28(3), 1387-1403.

2. The authors need to reconsider the significant figures in measurement for all length measurements shown in Figures 1, 3,4 and in text (ex. line 198). I question the validity of reporting to the 0.01 of a nm when using AFM to measure the lengths of these objects. For example, in line 198, the authors report a length of 219.43nm, I very much doubt the authors have this level of confidence or precision to measure to the 0.01nm. The authors should only report to the ones place.... thus for this case, 219 nm.

3. For diameter measurements, line 198, if the authors are using height measurements to assess the diameter of the CNCs, then 0.1 of a nm is likely valid. Thus in line 198, the reported value should be 8.0 and not 7.98.

4. Interesting to compare the "nanofiber" lengths reported in Figures 1f, 3e,f, and 4f,g, Which are ~100nm for WT, and 40-60nm for Osfc16.... to the CNC particle size measurements in Figure 6. It seems that the CNCs are of ~200nm for WT, and ~160nm for Osfc16, which are about 2 to 3 times larger in the size to the "nanofiber" features measured on the cell wall. The authors should give some description as to why this is the case.

5. I am a bit concerned with potential confusion of the terminology used in this paper of cellulose microfibrils (CMF) and cellulose nanofibers (CNF), as this is very similar to that used in the cellulose nanomaterials community, in which cellulose microfibrils (CMF) and cellulose nanofibrils (CNF) are isolated particles and defined by ISO TS 20477, and typically have branch-like structures. I am not sure if is possible for the authors to change their terminology, or at the very least it would be good for the authors to acknowledge that the CMF and CNF in the current work are still bonded to the plant cell wall, and will likely change morphology once they are extracted (by either mechanically or chemically process), and thus at this stage of the process of extraction should NOT to be considered comparable to cellulose microfibrils (CMF), and cellulose nanofibrils (CNF) used in the nanocellulose community. (something like this anyway). Actually, the authors demonstrated this by using acid hydrolysis to produce cellulose nanocrystals (CNCs), which are spindle shaped particles, a morphology completely different than CMF and CNF particle type used in the nanocellulose community.

6. Lines 97-104, and Figure 1b,e,f. The authors claim that they are imaging the localized attack or disintegration of the cellulose microfibrils that make up the cell wall. It is difficult to "see" the detailed features in the changes in cellulose microfibrils, Fig 1 b,e in which the image size is small, and the resolution is limited. Because of this it is challenging for the reader to definitively assess if the author claims are correct. It would be beneficial if the authors could add to the Supporting document a series of images and schematics that would help "walk" the reader through and show the detail AFM images and highlight the feature with arrows and labels of what features are changing. Also, for Figure 1,e OSfc16 at the 8% condition, the AFM image is of terribly low resolution that any meaningful feature comparison with the 16% condition is dubious at best.

7. If item #1 is address, then the results shown in figures 3 and 4 would be more reliable and convincing.

8. In general I found that figures 1,2,3,4, and 5, have too many parts, and each image and plots are too small to see meaningful information. This is all relevant information and should be in the paper, so

I am not sure what the best solution to this is. I zoomed in to see the details, so perhaps not that big of a deal.

Reviewer #1 (Remarks to the Author):

This manuscript reports research on a rice cellulose synthase mutant. The authors used AFM and other analytic methods to characterize the cell wall cellulose microfibril structure and nanocrystals generated from the rice straw, and found more amorphous regions of the cellulose microfibrils in the mutant than that in WT, which resulted in shorter nanocrystals after delignification process, and higher enzyme digestibility. This study is of interesting to understand the molecular mechanisms of biomass deconstruction process for bio-based products. There are some questions/concerns for the authors to consider.

Answer (A): Thank you so much for your encouraging comments. Below, we tried our best to answer all questions raised, and accordingly performed essential corrections in the revised manuscript as highlighted in RED.

1) Does the term "Single-molecular" in the title refer to cellulose? Cellulose is a bundle of beta-1,4-glucans (molecules) packed by inter- and intra-molecular interactions.

A: We presented "Single-molecular" in the title to highlight two novel single-molecular methods performed for distinct digestion of single-bunch of cellulose microfibril: (1) The AFM technique with CBM single-molecular recognition system was applied to directly observe the crystalline cellulose microfibrils that are interval by the amorphous cellulose regions accountable for cellulose nanofibers assembly; (2) Individual cellulase enzyme (EGII or CBHI+BG) as single-molecular chipper was applied for distinct digestion of single-bunch cellulose microfibril (Fig. 4a, b). To our knowledge, despite AFM single-molecular method being previously employed with cellulose, this study was the first time to observe the digestion of a single cellulose microfibril by single-molecular cellulase enzyme.

2) The authors use "microfibrils", "nanofibers" and "nanocrystals" for cellulose, it may be helpful to define these terms.

A: Thanks, we described those three definitions in the revised manuscript: please see lines 48-50 for "microfibrils", lines 67-68 for "nanocrystals", and lines 77-79 for "nanofibers".

3) It seems not clear to me what is "Classic and advanced AFM" (line 27).

A: The classic AFM refers to the AFM that only has the height/topography images, but the advanced AFM has both the height and single-molecular recognition images.

4) "high density of amorphous cellulose chains" (line 28), cellulose chain refers to beta-1,4-glucan, do the authors mean amorphous cellulose regions?

A: Yes, we changed the amorphous cellulose "chains" to amorphous cellulose "regions" to avoid any potential confusion.

5) "single-molecule probe" (line 68), does the functionalized AFM tip contains only one CBM molecule?

A: Not really, but only one CBM molecule "interacts" with cellulose, which was validated by specific curves in the Force-Distance test (Fig. 2e).

6) "native cellulose microfibrils" (line 78), are the microfibrils in the delignified cell wall still "native"?

A: The “native cellulose microfibrils” terminology is specifically defined for the cellulose microfibrils that are still bound with plant cell walls and also remain primary orientation *in situ*, which should thus be distinguished from completely separated/dispersed cellulose nanocrystals *in vitro*.

7) "native microfibrils ... at heating stage" (line 92-93), are these microfibrils still "native"?

A: The same as question #6 answered above.

8) "two defects/breakpoints derived from the amorphous cellulose chains" (lines 100-101), do these "defects/breakpoints" show higher or lower intensity in the AFM height images? why? how to confirm they are "amorphous cellulose chains"? not residue hemicelluloses or other matrix polymers that may be associated with cellulose surface, especially after NaClO₂ treatment.

A: Good comments. The "defects" show relatively lower intensity in the AFM height images, which means discontinuous breakpoints along cellulose microfibrils. Three steps were performed in this study as an evidence-chain to confirm the "defects" originated from the amorphous cellulose regions.

Step 1, 16% NaClO₂ extraction has exposed the defects along the cellulose microfibrils accountable for much reduced cellulose DP values examined, suggesting that the “defects” should be derived from the extraction/digestion of amorphous cellulose regions by the 16% NaClO₂ extraction performed.

Step 2, Endoglucanase (EG) is simply realized for initial digestion of amorphous cellulose regions rather than the crystalline cellulose, which has thus exposed similar density of defects to those of the 16% NaClO₂ extraction examined in this study.

Step 3, The *Osfc16* mutant has been previously characterized with the early termination of beta-1,4-glucan chain elongation for reduced cellulose DP value detected (Ref #17), which should be accountable for more defects/amorphous cellulose regions examined in this work.

In conclusion, three experimental observations provide evidence that is consist with the suggestion that the “defects” originate from the extraction/digestion of the amorphous cellulose regions.

In terms of hemicellulose involvement, we have preliminary data about trace xylan deep-entrapment with the amorphous cellulose regions, which may be one of causes for the amorphous cellulose formation. Given this is an interesting issue, it is being investigated in another project at our lab.

9) "be accounting for an increased density of ...mutant" (lines 103-104), is the observed decrease of nanofiber length caused by genetic modification in the mutant, or a result of chemical treatment?

A: As described above, because the *Osfc16* mutant is from the early termination of beta-1,4-glucan chain elongation for reduced cellulose DP value (Ref #17), “the increased density of... “should be mainly due to a genetic site-mutation of *OsCesA9* in the mutant, but the chemical treatment definitely aids to expose the “defects”/amorphous cellulose regions for nanofibers observation *in situ*.

10) "two defects/breakpoints" (line 124), are these broken cellulose chains? evidence?

A: As we answered for question #8, the defects should be the sites for initiating EG and/or CBHI enzyme digestion, which were thus defined as breakpoints for initiating and completing cellulose hydrolysis. To avoid any confusion, we deleted “/breakpoints” at line 124, but it is raised later (wherever it is explained in the text) in the revised manuscript.

11) "reducing ends (breakpoints" (lines 140-141), any evidence to confirm these are reducing ends in this study?

A: Broken cellulose microfibrils should expose both the reducing- and non-reducing ends, but the CBHI has been well characterized to be specific for initiating cellulose digestion from the reducing ends as described in the context of lines 155-157.

12) "to expose smoother and flatter surfaces" (line 145), what are those?

A: This is a description about the images observed on cell wall surfaces of the *Osf16* mutant after "CBHI+BG" digestion, which explained why it is different from the WT (Fig. 4b). Because the *Osf16* mutant was of length-reduced cellulose nanofiber assembly accountable for high-density defects/breakpoints/amorphous cellulose regions enabled for an efficient cellulose digestion, the images of mutant thus showed that the most nanofibers had been digested to expose a smooth and flat version as observed. To clarify this issue, we added more words (lines 157-161) to explain it in the revised manuscript.

13) "native amorphous cellulose chains act... chemical extraction" (lines 153-155), what are "native amorphous cellulose chains"? do these exist in both WT and the *Osf16* mutant? why were these "amorphous cellulose chains" not detectible before treatment (Fig 3c, 3d)?

A: As described above, alternating amorphous cellulose regions should exist in the native cell walls of both WT and *Osf16*, but the mutant should have high frequency/density of amorphous cellulose regions, due to its reduced cellulose DP from early termination of beta-1,4-glucan chain elongation (Ref #17). Because it has been reported that the original "amorphous cellulose regions" consist of only 4 glucose residues occurred every 300 glucose residues/crystalline cellulose chains (Ref #13), we think it should not well be detectable by the method of CBM3a probe recognition employed in this study. However, after 16% NaClO₂ extraction, this study has well exposed the amorphous regions and/or even broken chains for visualization.

14) "the N₂ adsorption/desorption method", provide reference for this method.

A: Thanks, we added it.

15) "nitric-acids extraction at 100C" (line 179), it seems not relevant with this study.

A: "Nitric-acetic acids extraction" is a classic method for crystalline cellulose extraction, which should digest/remove all amorphous cellulose regions/chains to obtain highly-crystalline cellulose substrates. As sharply reduced hexose yields were determined from enzymatic hydrolyses of these cellulose substrates, it provided the evidence that the amorphous cellulose regions are the breakpoints essential for initiation and completing of whole native cellulose microfibrils.

16) "By comparison... bioenergy crops" (lines 183-187"). This conclusion statement seems not clear, if "the intact cellulose microfibrils released much higher hexose yields...why the *Osf16* mutant that appeared to be more "defects" of cellulose microfibrils were more digestible?

A: Sorry for misunderstanding of this term. We defined the "intact cellulose" as whole native cellulose microfibrils consisting of amorphous (defect) and crystalline cellulose chains. As described above, the *Osf16* mutant was of high-density amorphous cellulose regions of whole native cellulose microfibrils,

and the 16% NaClO₂ extraction exposed the “defects” from the amorphous cellulose regions as the “breakpoints” for initiating enzymatic digestion of cellulose. To avoid any possible confusion, we clarified for the “amorphous cellulose regions”, “defects” and “breakpoints” in the revised manuscript according to these logic statements.

So, we also revised the “intact cellulose” as “whole cellulose microfibrils” in the revised manuscript.

17) "with 64% H₂SO₄" (line 191), it is not very clear why these are "highly-valuable bioproduction (line 193).

A: To our knowledge, different acid/alkali methods have been applied to generate CNCs substrates, and "with 64% H₂SO₄" being one of most effective methods for CNCs production as cited in reference #33. Meanwhile, it has been well reported that CNCs are the optimal intermediates for highly-valuable bioproduction.

18) "at 219.43nm length and 7.98 nm diameter" (lines 198-199), are these mean numbers? how are they correlated with Fig 3?

A: Yes, they are mean values acquired by measuring total randomly-selected 100 samples (n=100). For CNCs preparation, 64% H₂SO₄ process is a strong and non-specific chemical treatment that degrades a large part of the original crude cellulose to yield larger-size CNCs that are completely dispersed in the suspension, whereas the NaClO₂ process is a mild and specific treatment that mainly extracts lignin to expose cellulose nanofibers, and also to maintain the crystalline cellulose chains bound with cell walls. Despite it is hard to judge their correlation, they both should be accountable by average DP values of crude cellulose and crystalline cellulose substrates, suggesting cellulose biosynthesis is a major cause on the CNCs length and cellulose nanofibers length distinctive in the mutant and WT examined in this study.

19) "inner-broken cellulose chains" (line 203), any evidence?

A: This assumption/finding is mainly based on (1) the real-time observations of EG or CBHI+BG digestions *in situ* (Fig. 4a, b) and (2) significantly reduced CNC diameter (Fig. 6) in the *Osfc16* mutant. Therefore, without the inner-broken cellulose chains occurred in the mutants, we could not have observed any diverse short nanofibers (the inside of whole cellulose microfibrils) after 42 min EGII digestion or the smooth faces would not have been observed from the 48 min CBHI+BG digestion that should peel off all short nanofibers.

20) "for real-time observation" (line 217), any real-time data?

A: We defined the “time-course AFM images of CMFs digestion” as the “real-time AFM...” (Fig. 4 a, b).

21) "cellulose biosynthesis is one of major causes for amorphous cellulose formation in plant cell walls"(lines 233-234), is the amorphous cellulose exists only in the mutant caused by site-mutant in OsCESA9?

A: For alternating cellulose amorphous regions, they should exist in both WT and *Osfc16* mutant, which has been previously reported (Refs #17 & #18). But the mutant should have relatively higher frequency/density of amorphous cellulose regions as explained above.

22) "the inner-broken cellulose chains are also presumed in the *Osfc16* mutant"(lines 243-244), are the broken chains exist in the mutant, not in WT?

A: Based on the real-time observations of EG or CBHI+BG digestions *in situ* and thinner CNCs detected in the mutant (Fig. 4a, b) as described above, it is fairly clear about the inner-broken cellulose chain occurrence in the mutant, but it is hard to judge its existing in the WT. In other words, even though the WT has the inner-broken chains, its density should not be detectable from our current AFM technology.

23) "reduce cellulose ... best quality in the future" (lines 243-244), why reduced cellulose nanofibers are better? references?

A: In general, the decreased CNCs size should accompany by an increased surface-area-to-volume ratios and reducing ends, which should be more active and specific for its conversion into highly-valuable bioproducts from chemical process, as described in Refs #39 & #40 cited in the revised references. However, to our knowledge, the raised CNCs size may be of specific benefit for strengthening functional materials.

24) "Fig.1." (Line 587). The area showed in Fig. 1a appeared to be parenchyma cell walls, if all AFM images were taken from this area, they were likely primary cell walls. However, the *Osfc16* mutant is a site-mutant in a secondary cell wall cellulose synthase (*OsCESA9*, lines 232-233), how does it affect primary cell wall cellulose synthesis?

A: Good question. We initially chose the parenchyma-type secondary walls (pSWs), mainly due to its ease for location of AFM tips. To make sure of the wall type, we scanned the cell wall surface by AFM under 0% NaClO₂ (Fig. 1b), and the lignified walls were recognized for the following experiments. Meanwhile, the cell wall thickness of around 2 μm would be ideal pSWs for exploration. Below, we listed two published articles about the recognition of pSWs:

Please see:

1. Ding, S.-Y., Liu, Y.-S., Zeng, Y., Himmel, M.E., Baker, J.O. and Bayer, E.A. (2012) How does plant cell wall nanoscale architecture correlate with enzymatic digestibility? *Science* 338, 1055-1060.
2. Ding, S.-Y., Zhao, S. and Zeng, Y. (2014) Size, shape, and arrangement of native cellulose fibrils in maize cell walls. *Cellulose* 21, 863-871.

Thanks again for kind comments.

Reviewer #2 (Remarks to the Author):

In this manuscript, the authors make use of a rice mutant with reduced cellulose DP that they had previously characterised (Osfc16) to explore the impact of cellulose DP on cellulose fiber assembly and degradation. This manuscript requires professional editing for clarity, grammar and spelling throughout. It is hard to interpret the authors meaning in many places. Overall, the data lacks in depth analysis, particularly in relation to previous work. Strong statements are made that simply are not supported by the data.

Answer (A): Thanks for kind comments. Regarding the major comments, we realized that this study had raised several new terms such as “defect”/ “breakpoints”, “cellulose nanofibers”, “Inner-broken chains” and also performed novel experiments such as real-time observation of distinct digestions of individual cellulose microfibril by single enzyme (EGII) or CBH (+BG) in the mutant and WT, which may somehow lead to a difficulty for understanding of the major findings and novelty achieved in this work. However, we tried our best to revise it based on all comments raised by three reviewers. In addition, we had invited native co-author for editing English, and we did again in the revised manuscript.

1. In the introduction, some of the reviews cited are pretty old (~20 years) and I would recommend the authors identify some more recent summaries of the literature, as there has been much progress. Please remove ref#10, as there has been no subsequent evidence supporting a role for sitosterols in priming cellulose biosynthesis (in fact, this lack of evidence is discussed in reference #4).

A: Thanks. We updated the references list in the “Introduction” part as highlighted in RED in the revised manuscript.

2. Please introduce more information about the Osfc16 mutant and what is known about it from the previous publication. It’s core to the paper so “natural mutant” “defective at [sic] cellulose biosynthesis” is not sufficient.

A: We added a detailed description of the *Osfc16* mutant in lines 70-73.

3. Fig 1e – it is challenging to see the indicated break points – I would recommend including additional higher magnification inset panels so that it is clearer for the reader.

A: We re-organized all items in Fig. 1 to make each bigger and clearer, which was much better to expose the nanofibers and breakpoints in Fig 1e.

4. Fig 2 – in the text you mention an “untreated cell wall” control, but the figure says lignin. I would also be surprised that you don’t get any signal with untreated cell wall, as CBM3a does bind to cell wall (e.g. 10.1073/pnas.1005732107), although of course I would expect a weaker signal than in chlorite treated wall.

A: Sorry for the misleading about the control experiment.

1. We should have stated the recognition area first (Lower pictures in Fig. 2b-e). The dark regions in the recognition images indicate molecule interactions. Thus, the interactions happen only in some small regions in Fig. 2b&c but are not completely blank. Considering its much-low recognition efficiency and non-specificity, we thus selected curves in most area which happens with no interactions.

2. The “Untreated cell wall” surface should be covered by lignin and hemicellulose. To avoid any confusion, we replaced “the lignin” by “the untreated cell wall” in Fig 2.

5. If the data on classic AFM and CBM3a AFM is so well correlated (as the authors state), and if it also requires the chlorite treatment which removes almost all the lignin/hemicellulose, then what is the advantage?

A: The advantage of CBM3a AFM is to confirm the “crystalline” cellulose distribution, for which the classic AFM cannot make sure. Importantly, we added words (lines 168-178) and also discussed that the single molecular AFM approach may be more sensitive for measurement of short cellulose nanofibers.

So far, there are two hypotheses about the crystalline and amorphous cellulose distributions for whole native microfibrils assembly: one is about their alternating assembly along the microfibrils, and the other is about the crystalline cellulose chains covered by the amorphous ones. By using the CBM3a AFM, the experimental data suggests that the alternating hypothesis would be correct for WT and mutant (add words in lines 135-136), but the mutant contains inn-broken chains, which may be also defined as inside amorphous cellulose regions.

6. I would have preferred to see more discussion about what is know about cellobiohydrolase and b-glucosidase activity already. For example, the Igarashi et al. 2009, 2011 papers (10.1074/jbc.M109.034611; 10.1126/science.1208386), which used AFM to look at CBH activity are not cited.

A: Thanks, we added a discussion by citing two classic CBHI articles (Refs #29 and #30) as you recommend. Please see lines 157-161.

7. There are many statements that are simply not addressed by the data contained in this paper. For example, I don't see how you can really claim that the source of amorphous cellulose is "cellulose biosynthesis" based on the data in this paper. Yes, you have a CESA mutant with increased amorphous cellulose, but that does not indicate cause-effect, only correlation. I'm not saying it's not a possibility, I just don't see how your data supports this statement.

A: As we answered for Questions #13 & # 19 raised by Reviewer one above, the *Osfc16* mutant is a natural site-mutation of OsCESA9 isoform, and has been characterized with significantly reduced cellulose DP and length-reduced cellulose nanofibrils from the early termination of beat-1,4-glucan chain elongation (Refs #17 & #18). As OsCESA4, 7, 9 isoforms are equally functional for biosynthesis of beat-1,4-glucan chains (18-36 chains?) that form cellulose microfibrils, three isoforms are mostly likely to form heterotrimeric structure with symmetric CESA distribution for cellulose synthase complexes. So the site-mutation of OsCESA9 should cause a symmetric distribution of DP-reduced cellulose chains from the early termination of beat-1,4-glucan chain elongation, which should explain why the amorphous cellulose regions and inner-broken chains were raised in the mutant. In terms of this issues, we broadened the discussion (lines 260-268) in the revised manuscript to address this issue.

8. Abbreviations should be written out in full, as should species names e. g. line 66 should be *Oryza sativa* L. ssp japonica cv. Nipponbare (NPB); same for gene names.

There are also issues with reference formatting e.g. #3 should be O'Sullivan etc
A: We revised it.

Thanks again for all valuable comments above.

Reviewer #3 (Remarks to the Author):

Overall a fairly well written article, the approach, the mechanism, and the results are noteworthy.

Answer (A): Thanks a lot for your encouraging comments.

Some minor issues that authors should consider addressing to help with the clarity of the work.

1. For CNC particle size measurement, to remove any ambiguity, the authors state that they are using AFM height profiles to assess length and diameter of the particle. The authors could also consider cite the following paper or one similar to it that describes the approach used for CNC particle size measurement using AFM.

o Bushell, M., Meija, J., Chen, M., Batchelor, W., Browne, C., Cho, J. Y., ... & Johnston, L. J. (2021). Particle size distributions for cellulose nanocrystals measured by atomic force microscopy: An interlaboratory comparison. *Cellulose*, 28(3), 1387-1403.

A: Thanks, we cited this important article in the revised manuscript (Please see Ref # 49).

2. The authors need to reconsider the significant figures in measurement for all length measurements shown in Figures 1, 3,4 and in text (ex. line 198). I question the validity of reporting to the 0.01 of a nm when using AFM to measure the lengths of these objects. For example, in line 198, the authors report a length of 219.43nm, I very much doubt the authors have this level of confidence or precision to measure to the 0.01nm. The authors should only report to the ones place.... thus for this case, 219 nm.

A: We upgraded the measurement precision of lengths for all figures in the revised manuscript.

3. For diameter measurements, line 198, if the authors are using height measurements to assess the diameter of the CNCs, then 0.1 of a nm is likely valid. Thus in line 198, the reported value should be 8.0 and not 7.98.

A: We upgraded it in the revised manuscript.

4. Interesting to compare the “nanofiber” lengths reported in Figures 1f, 3e, f, and 4f, g, which are ~100nm for WT, and 40-60nm for Osfc16... to the CNC particle size measurements in Figure 6. It seems that the CNCs are of ~200nm for WT, and ~160nm for Osfc16, which are about 2 to 3 times larger in the size to the “nanofiber” features measured on the cell wall. The authors should give some description as to why this is the case.

A: Thanks for the good comments, because we had noticed these findings prior to the primary submission. Therefore, we added further details (lines 277-285) about these findings in the revised manuscript. The average length of nanofibers was scaled by the average distance of two amorphous cellulose regions on the surfaces of whole cellulose microfibrils after NaClO₂ extraction with lignin and small hemicelluloses. Based on our ongoing another project, the amorphous regions are determined by both cellulose biosynthesis and trace-hemicellulose (unextractable by NaClO₂) interaction. In terms of CNCs preparation, the strong 64% H₂SO₄ process should remove the surface of whole cellulose microfibrils and degrade small size nanocrystals, and thus CNCs formation is mainly derived from the inner long cellulose chains, which should explain why CNCs were of longer length compared to the cellulose nanofibers. On the other hands, it also explained why the mutant had

relatively shorter CNCs than those of the WT, due to much inner-broken cellulose chains occurred in the mutant as discussed above.

5. I am a bit concerned with potential confusion of the terminology used in this paper of cellulose microfibrils (CMF) and cellulose nanofibers (CNF), as this is very similar to that used in the cellulose nanomaterials community, in which cellulose microfibrils (CMF) and cellulose nanofibrils (CNF) are isolated particles and defined by ISO TS 20477, and typically have branch-like structures. I am not sure if is possible for the authors to change their terminology, or at the very least it would be good for the authors to acknowledge that the CMF and CNF in the current work are still bonded to the plant cell wall, and will likely change morphology once they are extracted (by either mechanically or chemically process), and thus at this stage of the process of extraction should NOT to be considered comparable to cellulose microfibrils (CMF), and cellulose nanofibrils (CNF) used in the nanocellulose community. (Something like this anyway). Actually, the authors demonstrated this by using acid hydrolysis to produce cellulose nanocrystals (CNCs), which are spindle shaped particles, a morphology completely different than CMF and CNF particle type used in the nanocellulose community.

A: Thanks for your concern. To avoid any confusion, we defined three terms in the revised manuscript: Please see lines 48-50 for “microfibrils”, lines 67-68 for “nanocrystals”, and lines 77-79 for “nanofibers”.

6. Lines 97-104, and Figure 1b,e,f. The authors claim that they are imaging the localized attack or disintegration of the cellulose microfibrils that make up the cell wall. It is difficult to “see” the detailed features in the changes in cellulose microfibrils, Fig 1 b,e in which the image size is small, and the resolution is limited. Because of this it is challenging for the reader to definitively assess if the author claims are correct. It would be beneficial if the authors could add to the Supporting document a series of images and schematics that would help “walk” the reader through and show the detail AFM images and highlight the feature with arrows and labels of what features are changing. Also, for Figure 1,e Osfc16 at the 8% condition, the AFM image is of terribly low resolution that any meaningful feature comparison with the 16% condition is dubious at best.

A: We addressed all these items in Fig. 1 to enlarge the size of each item so as to clearly illustrate exposing nanofibers and breakpoints.

7. If item #1 is address, then the results shown in figures 3 and 4 would be more reliable and convincing.

A: We addressed question#1 as your suggestion.

8. In general, I found that figures 1,2,3,4, and 5, have too many parts, and each image and plots are too small to see meaningful information. This is all relevant information and should be in the paper, so I am not sure what the best solution to this is. I zoomed in to see the details, so perhaps not that big of a deal.

A: Thanks. We reorganized all items in Figures 1, 2, 3, 4, and 5, and the imaging quality of all items seems much better.

Thank you again for all comments.

Reviewers' Comments:

Reviewer #1:

Remarks to the Author:

The revised manuscript has answered most of my questions. There are still concerns:

1. If the Osfc16 is a mutant of secondary cell wall cellulose synthase, is there experimental evidence to support its involvement in the synthesis of the parenchyma wall (thickened primary wall) in the stem ground tissue showed in all AFM images? In addition, some of the AFM images are not at the same scale, it would be helpful to show the same scan size to compare the differences between WT and Osfc16.
2. The results show that both the chemical treatment and genetic modification in CESA9 could affect cellulose structure, how could the authors suggest the increased density of "defects" in Osfc16 should be mainly due to a genetic site-mutation of OsCesA9? The authors assume NaClO₂ treatment "exposes" the "defects", could it possible the chemical treatment also "create" "defects"? see treatment of higher concentration of NaClO₂ results in lower DP (Fig. 5), or is it possible that the mutant plant has overall structural defects that increases the susceptibility of the cell walls to chemical treatment?
3. If we assume the fundamental structure of cellulose microfibril contains 18-36 chains, the AFM images of CMF, CNF, and CNC showed in this study appeared to be larger than the theoretical size of 18-36 chains microfibril, which seemed likely bundles of microfibrils (Fig. 2), thus the question is how these bundles align the crystalline and amorphous regions "breakpoints" to allow recognition by the CBM3a-probe at the nanometer scale?

Reviewer #2:

Remarks to the Author:

Thank you for providing detailed answers to my comments.

Clarity of manuscript - while there are still numerous typographical errors, the manuscript is significantly more readable. Definition of terms, description of materials etc all support the interpretation of the data by the reader.

Examples of errors e.g. line 74 "consequential lignin" should be "sequential lignin", line 85 should be "enzymatic hydrolyses of the size-reduced cellulose" etc. There are many, and I have not listed them.

I still think you should define "classic" and "advanced" AFM in the text, as I don't think these are particularly precise or future-proof terms.

Again, I find there is still a lack of context for some experiments. For example, what is known (and published by this group in PBJ!) about the Osfc16 mutant is still not included (although more detail than the first draft). Just describing what previous saccharification data shows for this line (and what pre-treatment/cellulase combo was used), and how these experiments build on this is useful. Same for how this is building on the previous development of CBM3a for use as a probe in AFM. It is important for the reader to know that this is not a first use in this article, but instead it is application of a developed method to a known mutant.

Please include stats in the figures when you report numbers e.g. the nanocrystal size should be +/- s.d.

Overall, this manuscript is much improved on the first submission. However, in my opinion, use of a single mutant line is not sufficient to merit publication in such a broad interest journal as this. I would recommend inclusion of different mutant lines in at least one set of AFM experiments, so that comparison can be achieved (e.g. hemicellulose vs cellulose in your signals).

Reviewer #3:

Remarks to the Author:

The authors have adequately address my comments on the paper.

Reviewer #1 (Remarks to the Author):

The revised manuscript has answered most of my questions. There are still concerns:

Answer (A): Thanks so much for kindly comments. Below, we tried our best to answer all questions raised.

1. If the *Osfc16* is a mutant of secondary cell wall cellulose synthase, is there experimental evidence to support its involvement in the synthesis of the parenchyma wall (thickened primary wall) in the stem ground tissue showed in all AFM images?

Answer (A): We realize that the reviewer is still concerned about whether the main findings achieved in this study were due to the defective cellulose biosynthesis for length-reduced nanofibers occurred only in the secondary cell walls of *Osfc16* mutant relative to its WT. Below, we should be convinced by the five evidences:

1. Before our AFM experiment *in situ*, we scanned the parenchyma-type secondary walls and parenchyma cell walls *in vivo* without any chemical extraction as shown below. As a comparison, we could observe the lignified surfaces in the parenchyma-type secondary cell walls, but cellulose microfibrils were clearly viewed in the typical primary cell walls.

2. In our recently-published article cited as Ref #18, we have used the stem tissues (typical secondary cell walls) of *Osfc16* mutant to observe length-reduced cellulose nanofibers distribution (Fig. 8 of the Ref) by performing high-pressure homogenization, and accordingly determined higher enzymatic hydrolysis of ground stem tissue *in vitro* relative to its WT, which should further confirm that the site-mutation of OsCESA9 should only cause a defective cellulose biosynthesis of secondary cell walls even though using another stronger processing approach for cellulose nanofibers preparation

3. In our just-accepted article (cited as Ref #39), we have generated a new site mutant of OsCESA9 by using CRISPR/Cas9 gene-editing. As a comparison with its WT (shown below), we have not only observed length-reduced cellulose nanofibers in the stem tissue *in situ* (Fig. 1D of the Ref), but have also determined higher biomass enzymatic saccharification of the ground stem tissues *in vitro*, re-confirming that the major findings should be derived from the defective cellulose biosynthesis of secondary cell walls in the new mutant.

4. More importantly, in our on-going project, we have performed classic protoplast cultures to observe typical creation of primary cell walls *in vitro* by using two distinct mutants: *Oscesa5* for cellulose biosynthesis of primary cell walls and *Osfc16* mutant for secondary cell walls in rice. As a comparison with their WT, the *Oscesa5* showed defective primary cell walls, whereas the *Osfc16* mutant did not exhibit any significantly altered primary cell wall formation, providing direct evidence that the major findings achieved in the *Osfc16* mutant should be only due to the defective cellulose biosynthesis of secondary cell walls.

5. In addition, we added a new experiment data to explore two other rice mutants *Osfc24* and *Osfc9*, which have been previously identified as defective cellulose biosynthesis in secondary cell walls. In those cell types, we still can observe their defective cellulose (Fig. S4). Thus, we think parenchyma-type secondary walls are suitable for secondary cell wall investigation.

Taken all together above, this study should provide consistent data about cellulose nanofiber assembly *in situ* and cellulose enzymatic hydrolysis *in vitro*, as well as cellulose nanocrystals formation, which should be only due to a defective cellulose biosynthesis of secondary cell walls

of *Osfc16* mutant examined. Therefore, we added more discussion in the revised manuscript as marked in RED.

2. In addition, some of the AFM images are not at the same scale, it would be helpful to show the same scan size to compare the differences between WT and *Osfc16*.

A: Thanks, we updated it in the revised manuscript.

3. The results show that both the chemical treatment and genetic modification in CESA9 could affect cellulose structure, how could the authors suggest the increased density of "defects" in *Osfc16* should be mainly due to a genetic site-mutation of *OsCesA9*? The authors assume NaClO_2 treatment "exposes" the "defects", could it possible the chemical treatment also "create" "defects"? see treatment of higher concentration of NaClO_2 results in lower DP (Fig. 5), or is it possible that the mutant plant has overall structural defects that increases the susceptibility of the cell walls to chemical treatment?

A: Good question. To our knowledge, plant cell wall structure is of extreme integrity and complex. To observe cellulose nanofiber assembly in native plant cell walls *in situ*, we had to perform appropriate chemical and enzymatic processes. Even though the NaClO_2 treatment has been examined for main extraction of lignin, the high NaClO_2 concentration seemed to create relatively shorter cellulose nanofibers *in situ*. So, it is possible to create new "defects" on crystalline regions rather than only exposes "defects" of the amorphous regions. However, despite it is hard to distinguish "create" and "expose", which depends on the degree of chemical and enzymatic processes, this study has found the similar defect frequency between the 16% NaClO_2 extraction and EG (endoglucanases) hydrolysis, leading to the 16% NaClO_2 process enough to expose amorphous regions. More importantly, as explained for Question #1 above, strong high-pressure homogenization could also lead to observation of length-reduced cellulose nanofibers accumulation in the *Osfc16* mutant, which should also be only due to a defective cellulose biosynthesis of secondary cell walls relative to its WT. So, once the *Osfc16* mutant and WT were processed under the exact same conditions, it could provide valid evidences to clarify for the major claims in this study.

4. If we assume the fundamental structure of cellulose microfibril contains 18-36 chains, the AFM images of CMF, CNF, and CNC showed in this study appeared to be larger than the theoretical size of 18-36 chains microfibril, which seemed likely bundles of microfibrils (Fig. 2), thus the question is how these bundles align the crystalline and amorphous regions "breakpoints" to allow recognition by the CBM3a-probe at the nanometer scale?

A: As AFM is a surface-probing technology, to our knowledge, it could detect the most surface signals along the CMFs. So, we are not very clear about what's the distribution or recognition signals inside/underneath the single or bundles of CMFs, which currently remains a technique difficulty to make sure they aligned or not aligned in the bundles. In other words, it would be an interesting and important study in the future, but should be dependent on the novel technology to explore such as *in vitro* synthesis of cellulose microfibrils by using native cellulose synthase complexes.

Thanks again for all comments above.

Reviewer #2 (Remarks to the Author):

Thank you for providing detailed answers to my comments.

Clarity of manuscript - while there are still numerous typographical errors, the manuscript is significantly more readable. Definition of terms, description of materials etc all support the interpretation of the data by the reader.

Answer (A): Thanks a lot for encouraging comments. We revised typographical errors, added some context, and updated figures as suggested. All corrections in the revised manuscript were marked in RED.

Examples of errors e.g. line 74 "consequential lignin" should be "sequential lignin", line 85 should be "enzymatic hydrolyses of the size-reduced cellulose" etc. There are many, and I have not listed them.

A: Thanks, we revised it.

I still think you should define "classic" and "advanced" AFM in the text, as I don't think these are particularly precise or future-proof terms.

A: Thanks, we added explanations in the revised manuscript (See Lines 84-87).

Again, I find there is still a lack of context for some experiments. For example, what is known (and published by this group in PBJ!) about the *Osf16* mutant is still not included (although more detail than the first draft). Just describing what previous saccharification data shows for this line (and what pre-treatment/cellulase combo was used), and how these experiments build on this is useful. Same for how this is building on the previous development of CBM3a for use as a probe in AFM. It is important for the reader to know that this is not a first use in this article, but instead it is application of a developed method to a known mutant.

A: Thanks, as we answered for the Question #1 raised by the Reviewer one above, we have published two articles about the *Osf16* mutants, and thus we added more context of *Osf16* mutant about its improvements on biomass enzymatic saccharification, Pickering emulsions productivity and lignocellulose-degradation enzymes secretion. We also added more information about the CBM3a probed AFM as suggested in the revised manuscript.

Please include stats in the figures when you report numbers e.g. the nanocrystal size should be +/- s.d.

A: Thanks, we added the stats in Figure 6.

Overall, this manuscript is much improved on the first submission. However, in my opinion, use of a single mutant line is not sufficient to merit publication in such a broad interest journal as this. I would recommend inclusion of different mutant lines in at least one set of AFM experiments, so that comparison can be achieved (e.g. hemicellulose vs cellulose in your signals).

A: Good suggestion. As we answered for the Question #1 raised by the Reviewer one above, we have just published an article (cited as Ref #39) about a new site mutant of *OscESA9* generated by classic CRISPR/Cas9 gene-editing. As a comparison with its WT, we have not only observed length-reduced cellulose nanofibers in the stem tissue *in situ*, but also determined higher biomass enzymatic saccharification of the ground stem tissues *in vitro*, consistent with the findings of *Osf16* mutant

reported in this manuscript. More importantly, our group has also generated a bench of site-mutants by CRISPR/Cas9 gene-editing of all OsCESA4,7,9 complexes essential for cellulose biosynthesis of secondary cell walls in rice, for which we are working on their diverse biological functions and multiple conversions for cost-effective biofuels and highly-valuable bioproducts under a green-like manner. In addition, we added new experiment data in this manuscript to explore two other rice mutants *Osfc24* and *Osfc9*, which have been previously identified as defective cellulose biosynthesis in secondary cell walls (Fig. S4). Interestingly, we could observe cellulose defects even though under 8% NaClO₂ extraction with those two mutants, suggesting that cellulose defects should be dependent on genetic engineering of cellulose biosynthesis.

To our knowledge, this study has provided basic knowledge and molecular mechanisms for in-depth understanding of the potential findings achieved in all new mutants in the future work. So, we added the new experiment in the supplementary (Fig. S4) and more discussions and perspective in the revised manuscript, which should strength the novelty and significance of this manuscript as concerned by the reviewer.

Thanks again for all valuable comments.

Reviewer #3 (Remarks to the Author):

The authors have adequately address my comments on the paper.

A: Thank you again for encouraging reply.

Reviewers' Comments:

Reviewer #1:

Remarks to the Author:

The revised manuscript is much improved. I don't have further comments.

Reviewer #2:

Remarks to the Author:

I thank the authors for continuing to work to address the comments.

The manuscript still requires editing for grammar and typographical errors.

I thank the authors for including data on additional mutants (Fig S4). However, please include this in the results not the discussion. Why do you think that *Osf9* and 24 show such clear result at 8% NaClO_2 , whereas *Osf16* is only at 16%?

Thank you also for including details of your new publication, which includes a new *cesa9* allele. Please can you comment on the differences in the crude cellulose DP observed between the two lines (*Osf16* and ref#39)? It seems substantially different. I had previously recommended using multiple independent mutant lines for this work, as is standard practice. This is because lines may carry additional mutations in other genes or regulatory elements that confound interpretation. If the phenotype observed is consistent across multiple independently generated lines, then we can conclude that it is due to this defect. Here you appear to have a significant difference between the two.

Other than that, I am happy that the authors have addressed all comments.

Reviewer #1 (Remarks to the Author):

The revised manuscript is much improved. I don't have further comments.

Answer (A): A: Thank you again for all your previous comments and this encouraging point.

Reviewer #2 (Remarks to the Author):

I thank the authors for continuing to work to address the comments.

Answer (A): Thank you so much for your kind comments. Below, we tried our best to answer the questions raised, and meanwhile, we completed all essential corrections in the revised manuscript as marked in RED.

The manuscript still requires editing for grammar and typographical errors.

A: We did seriously get through the manuscript to correct all mistakes in writing and grammar.

I thank the authors for including data on additional mutants (Fig S4). However, please include this in the results not the discussion. Why do you think that *Osfc9* and *24* show such clear result at 8% NaClO₂, whereas *Osfc16* is only at 16%?

A: Good suggestion. We shifted it to the “Results” section. As we have known, the *Osfc9* and *Osfc24* mutants are defective in the biosynthesis of CESA4, 7, 9 complexes, whereas the *Osfc16* mutant is only defective in the CESA9 isoform by site mutation, which should reflect the degree of defects in the cellulose synthesis complex and their cellulose microfibril production. It may also explain why the *Osfc9* and *Osfc24* mutants show clear defects at 8% NaClO₂ extraction, whereas the *Osfc16* mutant requires a 16% NaClO₂.

Thank you also for including details of your new publication, which includes a new *cesa9* allele. Please can you comment on the differences in the crude cellulose DP observed between the two lines (*Osfc16* and ref#39)? It seems substantially different. I had previously recommended using multiple independent mutant lines for this work, as is standard practice. This is because lines may carry additional mutations in other genes or regulatory elements that confound interpretation. If the phenotype observed is consistent across multiple independently generated lines, then we can conclude that it is due to this defect. Here you appear to have a significant difference between the two.

A: Good question. We guess that the reviewer suspected why the crude cellulose DP values measured in this study of WT and *Osfc16* are 730 vs 630, whereas in another study cellulose DP of WT and *cesa9* are 875 vs 750? To our current knowledge, the cellulose DP values (including cell wall compositions) are dynamically varied by multiple factors, and one major factor should be affected by the experimental materials harvested from different years. In fact, we have observed for many years that the cellulose DP values (including cell wall compositions) varied from the different harvest years, because the climate conditions could not be fixed from different years of field experiments. However, we remained to compare the mutants with their wild type from the same year of the field experiment.

Hence, in this study, the *Osfc16* and WT were harvested in the year of 2018, whereas the *cesa9* mutant and WT were harvested in the year of 2020. Although differences in absolute DP values exist, the defects of DP in the two mutants relative to WT are consistent. Moreover, we presented another data that the DP values of WT, *Osfc16*, and *cesa9s* are measured using the experimental materials harvested from the same year as shown below, and their DP defects relative to WT appear to be

consistent across different site mutant lines. However, in terms of the small DP variations among site-mutant lines, we think that they should reflect the different amino acids' site-mutations with different impacts on the CESA9 activity.

(cesa9-ZnF referred #39)

Other than that, I am happy that the authors have addressed all comments.

A: Thanks again for all the encouraging and valuable comments.

Reviewers' Comments:

Reviewer #2:

Remarks to the Author:

Thank you for the authors additional comments. I have nothing further to add.

REVIEWERS' COMMENTS

Reviewer #2 (Remarks to the Author):

Thank you for the authors additional comments. I have nothing further to add.

Answer (A): A: Thank you again for all your previous comments.